# Heat Driven Flows in Microsized Nematic Volumes: Computational Studies and Analysis

**Izabela Šliwa** [1,*] **and Alex V. Zakharov** [2,*]

1    Department of Mathematical Economics, Poznan University of Economics and Business,
     Al. Niepodleglosci 10, 61-875 Poznan, Poland
2    Saint Petersburg Institute for Machine Sciences, The Russian Academy of Sciences,
     199178 Saint Petersburg, Russia
*    Correspondence: izabela.sliwa@ue.poznan.pl (I.Š); alexandre.zakharov@yahoo.ca (A.V.Z.)

**Abstract:** The nematic fluid pumping mechanism responsible for the heat driven flow in microfluidic nematic channels and capillaries is described in a number of applications. This heat driven flow can be generated either by a laser beam focused inside the nematic microvolume and at the nematic channel boundary, or by inhomogeneous heating of the nematic channel or capillary boundaries. As an example, the scenario of the vortex flow excitation in microsized nematic volume, under the influence of a temperature gradient caused by the heat flux through the bounding surface of the channel, is described. In order to clarify the role of heat flux in the formation of the vortex flow in microsized nematic volume, a number of hydrodynamic regimes based on a nonlinear extension of the Ericksen–Leslie theory, supplemented by thermomechanical correction of the shear stress and Rayleigh dissipation function, as well as taking into account the entropy balance equation, are analyzed. It is shown that the features of the vortex flow are affected not only by the power of the laser radiation, but also by the duration of the energy injection into the microsized nematic channel.

**Keywords:** liquid crystals; hydrodynamic of isotropic systems; thermodynamics

## 1. Introduction

Impact of liquid crystal (LC) materials on modern technology is very impressive [1]. The recent technological revolution has been brought by these LC materials in the field of displays. By now, for instance, flat-panel LC displays (LCDs) are widely used in consumer electronics, personal computers, and mobile devices, as well as in many types of medical, transportation, and industrial equipment. With the further development of the LC-display market, the question then arises of what are the next areas of the LCs application? Perhaps there is no more appropriate directions for application of the LC materials than the LC sensors and actuators [2]. They have various advantages in comparison with other types of microsensors and microactuators; simple structure, high shape adaptability, easy downsizing, and low driving voltages. This is due to the fact that liquid crystals are extremely sensitive to external stimuli and therefore can be used for the construction of stimuli-responsive devices, such as sensors or actuators. There is another suitable direction for application of the LC materials, which is related to the precise manipulation of complex liquids on a microscale. The problem of electrically driven manipulation of complex liquids such as LCs has brought an increasing number of integrated small-scaled microdevices for chemical and biological applications. These are known as micrototal analysis systems or biological lab-on-a-chip systems [3–5]. Electro-osmosis, dielectrophoresis, and electrowetting have been explored for controlling microflows [3,4,6]. Thus, understanding the flow of LCs through micron-scale channels is of increasing significance. Nematic drops of appropriate size confined in the capillary are microdevices, whose molecular orientations also can be manipulated by a temperature gradient $\nabla T$, generated, for instance, by a laser beam focused both in the microfluidic volume or on the boundary of the LC channel. Fluid

pumping principle has been developed utilizing the interaction, on the one hand, between the gradient of director field $\nabla \hat{\mathbf{n}}$ and a temperature gradient $\nabla T$, excited, for instance, by a laser beam, focused both on the boundary of the LC channel or inside the LC microvolume, and, on the other hand, between the $\nabla \hat{\mathbf{n}}$ and the velocity field $\mathbf{v}$, excited in the microfluidic channel by the laser irradiation [7]. The problem of motion of an ultra-small (a few microliters) isotropic and LC drops, under the influence of the temperature gradient, caused by a laser beam, has drawn considerable attention [8–14]. Understanding of how the liquid crystal material can be manipulated under the influence of the temperature gradient is also a matter of great fundamental interest, as well as an essential part of knowledge in the field of soft materials science.

Despite the fact that the possibility of the formation of hydrodynamic flows in nematic channels under the influence of temperature gradients has been theoretically described since [15], only detailed numerical simulations performed within the framework of the extended Ericksen–Leslie theory [16,17] allowed us to recreate the complete picture of the formation of flows in nematic microchannels and capillaries. One of the aims of this review is to describe the various regimes of hydrodynamic flow formation due to the interaction between the temperature and director field gradients obtained by numerical modeling of these processes. This review is devoted to the latest results describing the possibilities of computational methods implemented in the framework of the nonlinear extension of the Ericksen–Leslie theory, with accounting the entropy balance equation [18]. Another purpose of our review is to show some routes in describing the laser-excited motion of nematics enclosed in a microsized channel with a free surface.

This is the first such review that describes in detail the role of thermomechanical forces in the formation of hydrodynamic flow in microsized nematic channels and capillaries. It is based on the nonlinear extension of the Ericksen–Leslie theory, supplemented by thermomechanical correction of the shear stress and Rayleigh dissipation function, and also takes into account the entropy balance equation. The fact that the main results were obtained by numerical methods indicates that experimenters still have a lot of work to do in order to create a more complete picture of the formation of hydrodynamic flows in microsized nematic channels and capillaries.

The layout of this article is as follows. In the next section, we give the review of theoretical background to the description of the physical mechanism responsible for the heat driven flow in the microfluidic rectangular nematic channel. The heat driven nematic flow in a cylindrical microfluidic channel is described in Section 3. Our conclusions are given in Section 4.

## 2. Heat Driven Nematic Flow in Rectangular Microfluidic Channel

We are primarily concerned with the description of the physical mechanism responsible for the heat driven flow in the microfluidic rectangular nematic channel. This heat driven flow $\mathbf{v}$, generated by the laser beam focused both on the restricted surface or inside the nematic microvolume can set up the temperature gradient [9,12,13]. A challenging problem in all such LC systems is the precise handling of nematic microvolume which requires self-contained micropumps of small volume flow (dynamic pumps). This the nematic pumping principle is based on the coupling between the velocity and director fields, together with accounting the effect of the temperature gradient. This problem will be treated in the framework of the extended Ericksen–Leslie theory [16,17], supplemented by the thermomechanical correction of shear stress [7,15], and the entropy balance [18] equation.

If a liquid crystal drop placed between two horizontal plates is at rest and if it is heated both from below or above, then due to the coupling of the director and temperature gradients, a thermomechanical force develops that can overcome the viscous, elastic, and anchoring forces, and the LC drop begins to move in the horizontal plane. Such horizontal motion of the LC drop due to uniform heating from below was first studied by Lehmann [19].

In the past decade, some research activity has been focused on the problem of formation the hydrodynamic flows in the microsized LC volumes excited by the thermomechanical force $\sigma_{zx}^{tm}$ [7,20–23]. In the case when the temperature gradient $\nabla T$ in the hybrid aligned nematic (HAN) volume is proportional to $\Delta T/d$, the hydrodynamic flow **v** excited by the temperature gradient is proportional to $v \sim \frac{d}{\eta}\sigma_{zx}^{tm}$, where $\Delta T = T_2 - T_1 > 0$ (the range $[T_2, T_1]$ falls within the stability region of the nematic phase) is the temperature difference on the LC volume boundaries, $\eta$ is the viscosity, and $\sigma_{zx}^{tm} \sim \xi\frac{\Delta T}{d^2}$ is the tangential component of the thermomechanical stress tensor $\sigma_{ij}^{tm}$, and $\xi$ is the thermomechanical constant. Any physical effect that induces flow in the LC volume leads to the coupling of that flow with the director field $\hat{\mathbf{n}}$. While having a subordinate role, the hydrodynamic flow **v** has been found to qualitatively change the orientational behavior of the director $\hat{\mathbf{n}}$ in temperature driven flow.

Therefore, we are primary concerned here with the description of the physical mechanism responsible for the flow in the LC channel confined between solid surfaces, which is excited by the temperature gradient $\nabla T$, and the magnitude of that flow is proportional to $\Delta T$, whereas the direction of **v** influences both the direction of the heat flux **q** and the character of the preferred anchoring of the average molecular direction $\hat{\mathbf{n}}$ to the restricted surfaces. We focused on the analysis of the problem of formation of hydrodynamic flows in microsized nematic channels under the influence of directed heat fluxes from the position of numerical methods. The analysis of the dynamics of Lehmann-type effects in chiral LCs is far from our research interests. For more information see, for instance, ref. [22].

In an attempt to better understand the dissipation processes in microfluidic nematic systems confined in the various microsized geometries, under the influences the temperature gradient, we will review a number of numerical studies of these nematic systems. With this aim, a number of theories which include the hydrodynamic equations describing both the director reorientation and flow of nematic fluid, as well as the redistribution of the temperature field, will be described. These equations also will be supplemented by appropriate anchoring conditions for the director field on the bounding surfaces, as well as the no-slip condition for the velocity field. Since the geometry of the microsized nematic channel or capillary affects the nature of the hydrodynamic flow that is formed, two cases will be considered. The first is when dealing with a rectangular channel, and second is the cylindrical capillary.

### 2.1. Formulation of the Balance of the Momentum and Torque Equations and Conductivity Equation for Nematic Fluids Confined in Rectangular Channel

The aim of this section is to show some useful routes for further examining of the validity of theoretical treatment of the orientational dynamics in the microsized rectangular nematic channel under the effect of externally directed heat flux **q** across the bounding surface. It will be done in the framework of the extended Ericksen–Leslie theory [16,17], supplemented by the thermomechanical correction of shear stress [7,15], and the entropy balance equation [18]. The first part of the section is devoted to the hydrodynamic model, which is the basis for describing the formation of flows in microsized hybrid aligned nematic (HAN) channels and capillaries under the action of the thermomechanical force. This force is the result of interaction of the director $\hat{\mathbf{n}}$ and temperature $T$ gradients, and it is responsible for the formation of the hydrodynamic flow **v** in these channels. In our case, the temperature gradient $\nabla T$ in the HAN channels is formed by focused laser radiation. In the second part of the section, a numerical analysis of two modes of heating the LC channels is given. These modes are characterized by different laser radiation power and duration: slow and fast heating modes caused by the laser irradiation focused on the lower boundary of the HAN channel.

With this aim we consider the microsized HAN channel delimited by two lower and upper horizontal solid surfaces, located at $z = 0$ and $z = d$, respectively, and two lateral solid surfaces at distance $L$ on scale on the order of micrometers. The coordinate system defined by this task assumes that the director $\hat{\mathbf{n}} = (n_x, 0, n_z) = \sin\theta\hat{\mathbf{i}} + \cos\theta\hat{\mathbf{k}}$, where

$\theta \equiv \theta(t, x, z)$ denotes the polar angle, i.e., the angle between the direction of the director $\hat{\mathbf{n}}$ and the normal $\hat{\mathbf{k}}$ to the horizontal boundary surfaces, is in the $XZ$ plane. Here, $\hat{\mathbf{i}}$ is the unit vector directed parallel to the horizontal boundaries of the HAN channel, and $\hat{\mathbf{j}} = \hat{\mathbf{k}} \times \hat{\mathbf{i}}$ (see Figure 1).

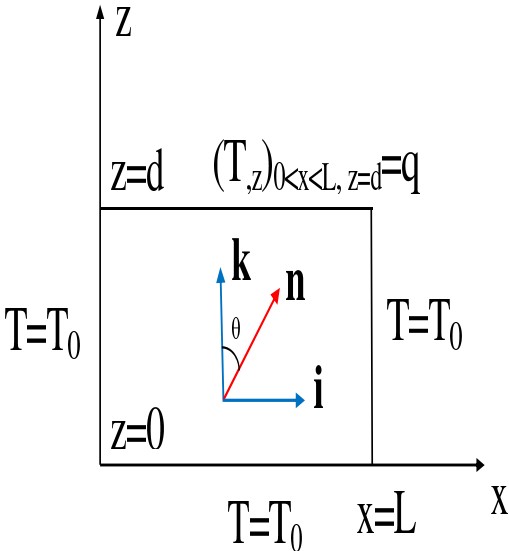

**Figure 1.** The coordinate system used for theoretical analysis. The $x$-axis is taken as being parallel to the director directions on the upper surface, $\theta(z, t)$ is the angle between the director $\hat{\mathbf{n}}$ and the unit vector $\hat{\mathbf{k}}$, respectively. Both the heat flux $\mathbf{q}$ and the unit vector $\hat{\mathbf{k}}$ are directed normal to the horizontal boundaries of the liquid crystal (LC) channel.

We shall be considering the HAN with the heat flux $\mathbf{q} = q\hat{\mathbf{k}}$ across the upper boundary

$$\lambda_\perp \left( \frac{\partial T(t, x, z)}{\partial z} \right)_{0 < x < L, z = d} = q, \tag{1}$$

whereas on the remaining boundaries the temperature is kept constant

$$T_{0 < x < L, z = 0} = T_{x = 0, 0 < z < d} = T_{x = L, 0 < z < d} = T_0. \tag{2}$$

Here $\lambda_\perp$ is the heat conductivity coefficient perpendicular to the director $\hat{\mathbf{n}}$. Therefore, the hybrid aligned nematic state, where at the upper boundary of the channel, the nematic phase is anchored homogeneously and the bulk of the nematic sample contains a gradient in $\theta(z) = \frac{\pi}{2d} z$ from homeotropic orientation at the lower surface to planar orientation at the upper surface, i.e.,

$$(n_x)_{x = 0, 0 < z < d} = (n_x)_{x = L, 0 < z < d} = 0, \ (n_x)_{0 < x < L, z = 0} = 0,$$
$$(n_x)_{0 < x < L, z = d} = 1, \tag{3}$$

or

$$\theta_{0 < x < L, z = 0} = \theta_{x = 0, 0 < z < d} = \theta_{x = L, 0 < z < d} = 0,$$
$$\theta_{0 < x < L, z = d} = \frac{\pi}{2}. \tag{4}$$

Taking into account that the width of the nematic channel $d \sim 1 - 5$ μm, one can assume that the mass density $\rho_m = $ const across the HAN channel, and one deals with an incompressible fluid. In turn, the incompressibility condition $\nabla \cdot \mathbf{v} = 0$ assumes that

$$u_{,x} + w_{,z} = 0, \tag{5}$$

where $u \equiv v_x(t, x, z)$ and $w \equiv v_z(t, x, z)$ are the components of the vector $\mathbf{v} = u\hat{\mathbf{i}} + w\hat{\mathbf{k}}$, and $u_{,\alpha} = \frac{\partial u}{\partial \alpha}$ ($\alpha = x, z$).

The hydrodynamic equations describing the reorientation of the nematic phase in $2D$ case, when the heat flux across the upper restricted surface exists, whereas the temperature on the rest surfaces is kept constant, can be derived from the balance of elastic, viscous, and thermomechanical torques $\mathbf{T}_{el} + \mathbf{T}_{vis} + \mathbf{T}_{tm} = 0$, the Navier-Stokes equation for the velocity field $\mathbf{v}$, excited by the heat flux $\mathbf{q}$, and the equation for the heat conduction.

The torque balance equation can be derived from the dimensionless balance of elastic $\mathbf{T}_{el} = \frac{\delta F_{el}}{\delta \hat{\mathbf{n}}} \times \hat{\mathbf{n}}$, viscous $\mathbf{T}_{vis} = \frac{\delta \mathcal{R}^{vis}}{\delta \hat{\mathbf{n}}_\tau} \times \hat{\mathbf{n}}$, and thermomechanical $\mathbf{T}_{tm} = \frac{\delta \mathcal{R}^{tm}}{\delta \hat{\mathbf{n}}_\tau} \times \hat{\mathbf{n}}$ torques [7,16,17], where $F_{el} = \frac{1}{2}\left[(\nabla \cdot \hat{\mathbf{n}})^2 + K(\hat{\mathbf{n}} \times \nabla \times \hat{\mathbf{n}})^2\right]$ is the dimensionless elastic energy, $K = K_3/K_1$, $K_1$ and $K_3$ are the splay and bend elastic constants of the nematic phase, $\hat{\mathbf{n}}_\tau \equiv \frac{d\hat{\mathbf{n}}}{d\tau}$ is the material derivative of $\hat{\mathbf{n}}$, whereas $2\gamma_1 \mathcal{R}^{vis} = \alpha_1(\hat{\mathbf{n}} \cdot \mathbf{B}_s \cdot \hat{\mathbf{n}})^2 + \gamma_1(\hat{\mathbf{n}}_\tau - \mathbf{B}_a \cdot \hat{\mathbf{n}})^2 + 2\gamma_2(\hat{\mathbf{n}}_\tau - \mathbf{B}_a \cdot \hat{\mathbf{n}}) \cdot (\mathbf{B}_s \cdot \hat{\mathbf{n}} - (\hat{\mathbf{n}} \cdot \mathbf{B}_s \cdot \hat{\mathbf{n}})\hat{\mathbf{n}}) + \alpha_4 \mathbf{B}_s : \mathbf{B}_s + (\alpha_5 + \alpha_6)(\hat{\mathbf{n}} \cdot \mathbf{B}_s \cdot \mathbf{B}_s \cdot \hat{\mathbf{n}})^2$ is the viscous, and $\delta_1 \mathcal{R}^{tm} = (\hat{\mathbf{n}} \cdot \nabla \chi)\mathbf{B}_s : \mathbf{A} + \nabla \chi \cdot \mathbf{B}_s \cdot \mathbf{A} \cdot \hat{\mathbf{n}} + (\hat{\mathbf{n}} \cdot \nabla \chi)(\hat{\mathbf{n}}_\tau - \mathbf{B}_a \cdot \hat{\mathbf{n}} - 3\mathbf{B}_s \cdot \hat{\mathbf{n}} + 3(\hat{\mathbf{n}} \cdot \mathbf{B}_s \cdot \hat{\mathbf{n}})\hat{\mathbf{n}}) \cdot \mathbf{A} \cdot \hat{\mathbf{n}} + \hat{\mathbf{n}}(\nabla \mathbf{v})^T \cdot \mathbf{A} \cdot \nabla \chi + \frac{1}{2}(\hat{\mathbf{n}} \cdot \mathbf{B}_s \cdot \hat{\mathbf{n}})\nabla \chi \cdot \mathbf{A} \cdot \hat{\mathbf{n}} + \hat{\mathbf{n}}_\tau \cdot \mathbf{A} \cdot \nabla \chi + \frac{1}{2}\mathcal{A}_0 \nabla \chi \cdot \nabla \mathbf{v} \cdot \hat{\mathbf{n}} + (\hat{\mathbf{n}} \cdot \nabla \chi)\mathcal{A}_0(\hat{\mathbf{n}} \cdot \mathbf{B}_s \cdot \hat{\mathbf{n}}) + \frac{1}{2}\mathcal{A}_0 \hat{\mathbf{n}}_\tau \cdot \nabla \chi$ is the thermomechanical contributions to the full dimensionless Rayleigh dissipation function [7,15–17], respectively. Here $\mathbf{B}_s = \frac{1}{2}\left[\nabla \mathbf{v} + (\nabla \mathbf{v})^T\right]$ and $\mathbf{B}_a = \frac{1}{2}\left[\nabla \mathbf{v} - (\nabla \mathbf{v})^T\right]$ are the symmetric and asymmetric contributions to the rate of strain tensor, $\mathbf{A} = \frac{1}{2}\left[\nabla \hat{\mathbf{n}} + (\nabla \hat{\mathbf{n}})^T\right]$, $\mathcal{A}_0 = \nabla \cdot \hat{\mathbf{n}}$ is the scalar invariant of the tensor $\mathbf{A}$, $(\nabla \hat{\mathbf{n}})^T$ is the transposition of $\nabla \hat{\mathbf{n}}$, and $\alpha_i$ ($i = 1 \div 6$) are the Leslie viscosity coefficients, respectively. We use here the invariant, multiple dot convention: $\mathbf{ab} = a_i b_j$, $\mathbf{a} \cdot \mathbf{b} = a_i b_i$, $\mathbf{A} \cdot \mathbf{B} = A_{ik} B_{kj}$, and $\mathbf{A} : \mathbf{B} = A_{ik} B_{ki}$, where repeated Cartesian indices are summed.

To be able to observe the formation of the hydrodynamic flow $\mathbf{v}$ in the HAN channel, under the effect of the heat flux $\mathbf{q}$, when the heating occurs during some times $\tau_{in}$, let us consider the dimensionless analog of the torque balance equation [7,16,17,21]

$$\begin{aligned} n_z n_{x,\tau} - n_x n_{z,\tau} = &\, \delta_1 \left[n_z \mathcal{A}_{0,x} - n_x \mathcal{A}_{0,z} - K(n_z h_z + n_x h_x)\right] \\ &- \frac{1}{2}\psi_{,xx}\left[1 + \gamma\left(n_x^2 - n_z^2\right)\right] - \frac{1}{2}\psi_{,zz}\left[1 - \gamma\left(n_x^2 - n_z^2\right)\right] \\ &\, 2\gamma\psi_{,xz}n_x n_z + \psi_{,z}\mathcal{N}_x + \mathcal{N}_z \psi_{,x} + \delta_2(\chi_{,x}\mathcal{L}_{,x} + \chi_{,z}\mathcal{L}_{,z}), \end{aligned} \tag{6}$$

where $\tau = t/t_T$ is the dimensionless time, $t_T = \rho C_p d^2 / \lambda_\perp$, $C_p$ is the heat capacity, $\gamma = \gamma_2/\gamma_1$, $\gamma_1$ and $\gamma_2$ are the rotational viscosity coefficients, $\mathcal{A}_0 = n_{x,x} + n_{z,z}$, $\chi \equiv \chi(\tau, x, z) = T(\tau, x, z)/T_{NI}$ is the dimensionless temperature, $T_{NI}$ is the nematic-isotropic transition temperature, $\chi_{,\alpha} = \frac{\partial \chi}{\partial \alpha}$ ($\alpha = x, z$), $\bar{z} = \frac{z}{d}$ is the dimensionless distance away from the lower boundary of the HAN channel, $\bar{x} = \frac{x}{L}$ is the dimensionless space variable corresponding to $x$-axis, respectively, $\bar{\psi} = \frac{t_T}{d^2}\psi$ is the scaled analog of the stream function $\psi$ for the velocity field $\mathbf{v} = u\hat{\mathbf{i}} + w\hat{\mathbf{k}} = -\nabla \times \hat{\mathbf{j}}\psi$ (see the ref. [21]), $n_{z,\tau} = \frac{\partial n_z}{\partial \tau}$, $\mathcal{N}_x = n_x n_{z,x} - n_z n_{x,x}$, $\mathcal{N}_z = n_z n_{x,z} - n_x n_{z,z}$, $\mathcal{L}_x = n_x n_{z,x} - \frac{3}{2}n_z n_{x,x} + \frac{1}{2}n_x n_{x,z}$, and $\mathcal{L}_z = -n_z n_{x,z} + \frac{3}{2}n_x n_{z,z} - \frac{1}{2}n_z n_{z,x}$, respectively. Here, $\delta_1 = \frac{t_T K_1}{\gamma_1 d^2}$ and $\delta_2 = \frac{\rho C_p T_{NI}}{\lambda_\perp}\frac{\xi}{\gamma_1}$ are two parameters of the nematic system, and $\xi$ is the thermomechanical constant [7,15].

The dimensionless Navier–Stokes equation for the velocity field $\mathbf{v} = u\hat{\mathbf{i}} + w\hat{\mathbf{k}} = -\nabla \times \hat{\mathbf{j}}\psi$ takes the form [21]

$$\begin{aligned} \delta_3 \psi_{,xz\tau} = &\, a_1 \psi_{,zzzz} + a_2 \psi_{,xzzz} + a_3 \psi_{,xxzz} + a_4 \psi_{,xxxz} + a_5 \psi_{,xxxx} + a_6 \psi_{,zzz} + \\ &\, a_7 \psi_{,xzz} + a_8 \psi_{,xxz} + a_9 \psi_{,xxx} + a_{10} \psi_{,zz} + a_{11} \psi_{,xz} + a_{12} \psi_{,xx} + \mathcal{F}, \end{aligned} \tag{7}$$

whereas the dimensionless entropy balance can be written as [21]

$$
\chi_{,\tau} = \left[ \chi_{,x} \left( \lambda n_x^2 + n_z^2 \right) + (\lambda - 1) n_x n_z \chi_{,z} \right]_{,x} +
$$
$$
\left[ \chi_{,z} \left( \lambda n_z^2 + n_x^2 \right) + (\lambda - 1) n_x n_z \chi_{,x} \right]_{,z} +
$$
$$
\delta_4 \chi \left( \nabla \cdot \frac{\partial \mathcal{R}^{tm}}{\partial \nabla \chi} \right) - \psi_{,z} \chi_{,x} + \psi_{,x} \chi_{,z}, \tag{8}
$$

where $\lambda = \lambda_\parallel / \lambda_\perp$ is the dimensionless parameter, $\lambda_\parallel$ is the heat conductivity coefficient parallel to the director. It should be noted that the function $\mathcal{F} = \left( \sigma_{xx}^{el} + \sigma_{xx}^{tm} - \sigma_{zz}^{el} - \sigma_{zz}^{tm} \right)_{,xz} + \left( \sigma_{zx}^{el} + \sigma_{zx}^{tm} \right)_{,zz} - \left( \sigma_{xz}^{el} + \sigma_{xz}^{tm} \right)_{,xx}$, the coefficients $a_i$ $(i = 1, \ldots, 12)$, the functions $\sigma_{ij}^{tm}$ $(i, j = x, z)$ and $\sigma_{ij}^{el}$ $(i, j = x, z)$ are given in the ref. [21], whereas $\sigma_{ij} = \mathcal{P} \delta_{ij} + \sigma_{ij}^{el} + \sigma_{ij}^{vis} + \sigma_{ij}^{tm}$ $(i, j = x, z)$ is the stress tensor (ST) of the nematic system, $\mathcal{P}$ is the hydrostatic pressure in the HAN channel, and $\sigma_{ij}^{el}$, $\sigma_{ij}^{vis}$, and $\sigma_{ij}^{tm}$ are the dimensionless ST components corresponding to the elastic, viscous, and thermomechanical forces, respectively. The set of the rest parameters of the nematic system, corresponding to the case of $4 - cyano - 4' - pentylbiphenyl$ ($5CB$), are: $\delta_3 = \frac{\rho d^2}{\gamma_1 t_T}$ and $\delta_4 = \frac{\xi}{\lambda_\perp t_T}$, respectively.

In the following we are focused primary on the heat conduction regime in the HAN channel which assumes that across the upper surface the heat flux is set up (see Equation (1)), whereas on the rest surfaces the temperature is kept constant (see Equation (2)). Physically, this means that across the LC channel the temperature gradient $\nabla T$, directed from the cooler to the warmer surfaces, excited by the heat flux **q** across to the upper boundary, may be built up.

Notice that the dimensionless ST $\sigma_{ij}$ can be obtained directly from the elastic contribution to the energy and Rayleigh dissipation function as $\sigma^{el} = -\frac{\partial F_{el}}{\partial \nabla \hat{\mathbf{n}}} \cdot (\nabla \hat{\mathbf{n}})^T$, $\sigma^{vis} = \frac{\delta \mathcal{R}^{vis}}{\delta \nabla \mathbf{v}}$, and $\sigma^{tm} = \frac{\delta \mathcal{R}^{tm}}{\delta \nabla \mathbf{v}}$, for the elastic, viscous, and thermomechanical contributions, respectively.

Straightforward calculations give the following expressions for ST components $\sigma_{ij}^{el}$, $\sigma_{ij}^{vis}$, and $\sigma_{ij}^{tm}$ which are listed in the ref. [21].

Now the evolution of the director in the HAN channel confined between two horizontal and two vertical solid surfaces, under the influence of viscous, elastic, and thermomechanical forces and taking into account the flow, can be obtained by solving the system of the nonlinear partial differential Equations (6)–(8) with the appropriate dimensionless boundary conditions for the director $\hat{\mathbf{n}}$ or the polar angle $\theta$

$$
(n_x)_{x=0,0<z<1} = (n_x)_{x=1,0<z<1} = (n_x)_{0<x<1,z=0} = 0, \ (n_x)_{0<x<1,z=1} = 1,
$$
$$
\theta_{0<x<1,z=0} = \theta_{x=0,0<z<1} = \theta_{x=1,0<z<1} = 0,
$$
$$
\theta_{0<x<1,z=1} = \frac{\pi}{2}, \tag{9}
$$

velocity field $\mathbf{v} = u\hat{\mathbf{i}} + w\hat{\mathbf{k}} = -\nabla \times \hat{\mathbf{j}} \psi$

$$
u_{0<x<1,z=0} = (\psi_x)_{0<x<1,z=0} = u_{x=0,0\le z<1} = (\psi_x)_{x=0,0\le z<1} = 0,
$$
$$
u_{x=d,0\le z<1} = (\psi_x)_{0\le x\le 1,z=1} = u_{0\le x\le 1,z=1} = (\psi_x)_{0\le x\le 1,z=1} = 0,
$$
$$
w_{0<x<1,z=0} = (\psi_z)_{0<x<1,z=0} = w_{x=0,0\le z<1} = (\psi_z)_{x=0,0\le z<1} = 0,
$$
$$
w_{x=d,0\le z<1} = (\psi_z)_{x=d,0\le z<1} = w_{0\le x\le 1,z=1} = (\psi_z)_{0\le x\le 1,z=1} = 0, \tag{10}
$$

temperature field $\chi(\tau, x, z)$

$$
\chi_{0\le x\le 1,z=0} = \chi_{x=0,0<z<1} = \chi_{x=1,0<z<1} = \chi_0,
$$
$$
\left( \frac{\partial \chi(\tau, x, z)}{\partial z} \right)_{0\le x\le 1,z=1} = Q, \tag{11}
$$

and the initial condition taken in the form

$$\hat{\mathbf{n}}(\tau = 0, x, z) = \hat{\mathbf{n}}^{eq}_{elast}(x, z), \tag{12}$$

respectively. Here, $\chi_0 = \frac{T_0}{T_{NI}}$, and $Q = -\frac{d}{T_{NI}\lambda_\perp} q$ is the dimensionless heat flux across the upper restricted surface. The laser-induced heating was used to inject energy $Q$ across the bounding surface at the microscopic scale [8,13]. In that case the nematic channel was heated by a laser beam, focused, for instance, on the upper bounding surface ($z = 1, x = x_0$), with intensity $\mathcal{I}(x) = \frac{2P_0}{\pi\omega_0^2} \exp\left(-\frac{2(x-x_0)^2}{\omega_0^2}\right)$, where $P_0$ is the laser power, and $\omega_0$ is the Gaussian spot size. Taking into account that the total absorbed laser power is $P_{in} = \alpha\mathcal{I}(x)$, the heat flux across the upper restricted surface can be written as

$$Q = Q_0 \exp\left(-\frac{2(x-x_0)^2}{\overline{\omega}_0^2}\right) \mathcal{H}[\tau_{in} - \tau], \tag{13}$$

where $\alpha$ is the absorption coefficient, $\overline{\omega}_0 = \frac{\omega_0}{d}$, $Q_0 = -\frac{2\alpha P_0 d}{\pi\omega_0^2 \lambda_\perp T_{NI}}$ is the dimensionless heat flux's coefficient, $\mathcal{H}[\tau_{in} - \tau]$ is the Heaviside step function, and $\tau_{in}$ is the duration of the energy injection into the nematic sample. Here, and everywhere else in this section, $Q_0$ is equal to $|Q_0|$.

For the case of $4 - cyano - 4' - pentylbiphenyl$ ($5CB$), at temperature corresponding to nematic phase, the first four parameters $\delta_1 = \frac{\rho C_p K_1}{\lambda_\perp \gamma_1}$, $\delta_2 = \frac{\rho C_p T_{NI}\xi}{\lambda_\perp \gamma_1}$, $\delta_3 = \frac{\lambda_\perp}{\gamma_1 C_p}$, and $\delta_4 = \frac{\xi}{\rho C_p d^2}$, that are involved in Equations (6)–(8) have the following values [21]: $\delta_1 \sim 10^{-3}$, $\delta_2 \sim 0.3$, $\delta_3 \sim 10^{-6}$, and $\delta_4 \sim 10^{-4}$.

Using the fact that $\delta_3 \ll 1$, the Equation (7) can be considerably simplified and takes the form [21]

$$a_1\psi_{,zzzz} + a_2\psi_{,xzzz} + a_3\psi_{,xxzz} + a_4\psi_{,xxxz} + a_5\psi_{,xxxx} + a_6\psi_{,zzz} +$$
$$a_7\psi_{,xzz} + a_8\psi_{,xxz} + a_9\psi_{,xxx} + a_{10}\psi_{,zz} + a_{11}\psi_{,xz} + a_{12}\psi_{,zz} + \mathcal{F} = 0, \tag{14}$$

where $a_i$ ($i = 1, \ldots, 12$) and $\mathcal{F}$ are functions which have been defined in ref. [21].

*2.2. Orientational Relaxation of the Director, Velocity, and Temperature Fields in Rectangular Microsized Nematic Channels*

With the evolution of the director $\hat{\mathbf{n}}$ to its equilibrium orientation $\hat{\mathbf{n}}_{eq}$, which is described by the polar angle $\theta(\tau, x, z)$ from the initial $\theta(\tau = 0, x, z) = \theta_{elast}(x, z)$ to the equilibrium $\theta_{eq}(x, z)$ conditions, both the horizontal and vertical components of the velocity field $\mathbf{v} = u\hat{\mathbf{i}} + w\hat{\mathbf{k}} = -\nabla \times \hat{\mathbf{j}}\psi$, and the temperature field $\chi(\tau, x, z)$ redistribution, can be obtained by solving the system of the nonlinear partial differential Equations (6), (8) and (14), together with the boundary conditions (9)–(11), and the initial condition taken in the form of (12). The calculations have been carried out by using both the relaxation [24] and the sweep [25] methods. The initial distribution of the director field $\hat{\mathbf{n}}_{el}(x, z)$ has been obtained from Equation (6) by means of the relaxation method with $\psi_{,x} = \psi_{,z} = (\chi)_{,x} = (\chi)_{,z} = 0$, and with the boundary $(n_x)_{x=0,0\le z\le 1} = (n_x)_{x=1,0\le z\le 1} = (n_x)_{z=1,0\le x\le 1} = 0$, $(n_x)_{z=0,0\le x\le 1} = 1$, and initial $n_x = \frac{1-z}{2}$, for $0 < x < 1$, and $n_x = 0$, for $x = 0, 1$, conditions. Having obtained the initial distribution of the director field $\hat{\mathbf{n}}_{el}(x, z)$, the initial distribution of the temperature field $\chi(\Delta\tau, x, z)$, corresponding to the first time step $\Delta\tau$, has been obtained from Equation (8), by means of the relaxation method with $\psi_{,x} = \psi_{,z} = 0$ and with the boundary and initial conditions in the form of Equations (9)–(12). Then the initial distribution of the stream function $\psi(\Delta\tau, x, z)$, corresponding to the first time step $\Delta\tau$ can be calculated by using the initial distributions of the director $\hat{\mathbf{n}}_{el}(x, z)$ and temperature $\chi(\Delta\tau, x, z)$ fields, as well as the function $\mathcal{F}$, which is involved in Equation (14). The next time step $\Delta\tau$ for the velocity and temperature fields, as well as for the director's distribution

across the nematic sample is initiated by the sweep method. The stability of the numerical procedure for Equations (6), (8) and (14) was defined by the following conditions [24]:

$$\frac{\Delta\tau}{\delta_3}\left(\frac{1}{(\Delta x)^2}+\frac{1}{(\Delta z)^2}\right)\le\frac{1}{2},$$

$$\frac{3a_5}{(\Delta x)^4}-\frac{2a_1}{(\Delta z)^4}>0,$$

where $\Delta x$ and $\Delta z$ are the space steps in the $x$ and $z$ directions, and $a_1$ and $a_5$ are the coefficients defined in Equation (14). In the calculations, the relaxation criterion $\epsilon = |(\chi_{(m+1)}(\tau,x,z)-\chi_{(m)}(\tau,x,z))/\chi_{(m)}(\tau,x,z)|$ was chosen to be equal to $10^{-4}$, and the numerical procedure was then carried out until a prescribed accuracy has been achieved. Here, $m$ is the iteration number and $\tau_R$ is the relaxation time of the system.

A. Slow heating mode

The evolution of the polar angle $\theta(\tau,x=0.5,z)$ in the middle part of the rectangular HAN channel ($x=0.5$) to its equilibrium distribution $\theta_{eq}(x=0.5,z)=\theta(\tau=\tau_6,x=0.5,z)$, during the slow heating mode, under the effect of the dimensionless heat flux $Q=0.44$ ($\sim 3.7\times 10^{-3}$ mW/$\mu$m$^2$), caused by the laser irradiation focused on the upper boundary of the HAN channel, at different times, from $\tau_1=0.00042$ ($\sim 54$ $\mu$s) [curve (1)] to $\tau_6=\tau_{in}=0.0134$ [curve (6)], respectively, is shown in Figure 2a. Figure 2b shows the same as in Figure 2a, but at different times, from $\tau_7=0.014$ [curve (7)] to $\tau_{11}=\tau_R=0.22$ [curve (11)], corresponding to the cooling mode. Here, the following times were used $\tau_{i+1}=2^i\tau_1$, $i=1,\ldots,5$, whose values increase from curve (1) to curve (6), and $\tau_1=0.00042$, and times $\tau_{i+1}=2^{i-6}\tau_7$, $i=7,\ldots,10$, whose values increase from curve (7) to curve (11), and $\tau_7=0.014$, respectively, whereas the time $\tau_R$ denotes the relaxation time of the LC system, and $\tau_{in}\sim\tau_6\sim 0.0134$ ($\sim 1.9$ ms) is the duration of the energy injection into the HAN channel across the upper boundary by the infrared laser with the power $P_0=14.3$ mW. The distance $z$ dependence of the polar angle $\theta(\tau,x=0.5,z)$ is characterized by the monotonic increase $\theta$ from $\theta(x=0.5,z=0)=0$, on the lower cooler boundary of the HAN channel, to $\theta(x=0.5,z=1)=\frac{\pi}{2}$, on the upper warmer boundary, respectively. The evolution of the polar angle $\theta(\tau,x,z=0.97)$ to its equilibrium distribution $\theta_{eq}(x,z=0.97)=\theta(\tau=\tau_6,x,z=0.97)$ along the width of the HAN channel ($0\le x\le 1$) in the vicinity of the upper warmer boundary of the channel ($z=0.97$), during the slow heating mode at different times, from $\tau_1=0.00042$ ($\sim 54$ $\mu$s) [curve (1)] to $\tau_6=\tau_{in}=0.0134$ [curve (6)], is shown in Figure 3a, whereas the cooling mode, at different times, from $\tau_7=0.014$ [curve (7)] to $\tau_{11}=\tau_R=0.22$ [curve (11)], is shown in Figure 3b, respectively. Both the slow heating (see Figure 3a) and cooling (see Figure 3b) modes are characterized by nonsymmetric profile of $\theta(\tau,x,z=0.97)$ with respect to the middle part ($x=0.5$) of the HAN channel, which is caused by nonsymmetric effect of the velocity field components $u(\tau,x,z)$ (see Figure 4a,b and Figure 5a,b) and $w(\tau,x,z)$ (see Figure 6a,b and Figure 7a,b), respectively.

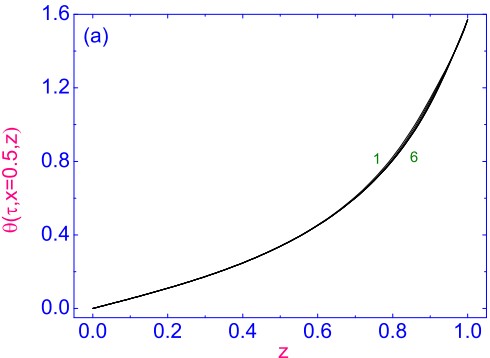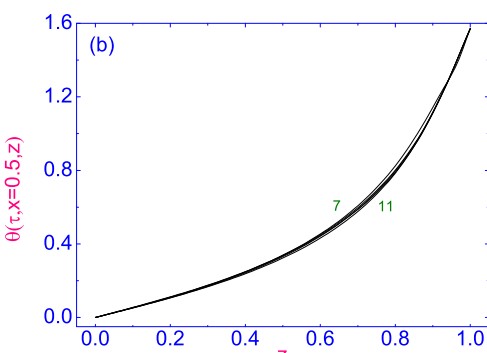

**Figure 2.** (**a**) The distance $z$ dependence of the polar angle $\theta(\tau, x = 0.5, z)$ (in rad) during its evolution to the equilibrium distribution $\theta_{eq}(x = 0.5, z) = \theta(\tau = \tau_6, x = 0.5, z)$ in the middle part of the hybrid aligned nematic (HAN) channel [21], corresponding to the slow heating mode is given starting from $\tau_1 = 0.00042$ [curve (1)] to $\tau_6 = \tau_{in} = 0.0134$ [curve (6)], respectively. (**b**) The same as in (**a**), but at different times, from $\tau_7 = 0.014$ [curve (7)] to $\tau_{11} = \tau_R = 0.22$ [curve (11)], corresponding to the cooling mode.

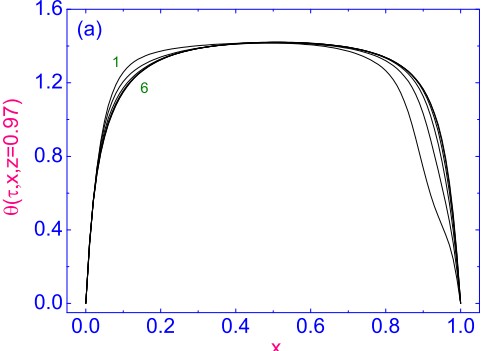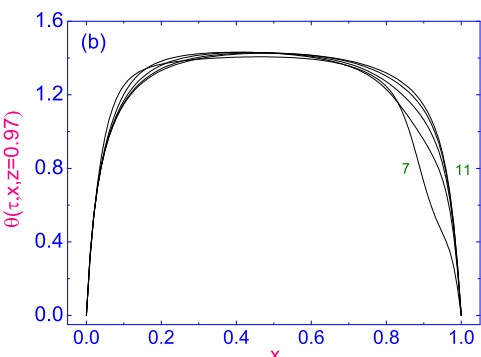

**Figure 3.** (**a**) The distance $x$ dependence of the polar angle $\theta(\tau, x, z = 0.97)$ (in rad) during its evolution to the equilibrium distribution $\theta_{eq}(x, z = 0.97) = \theta(\tau = \tau_6, x, z = 0.97)$ [21], along the width of the HAN channel ($0 \leq x \leq 1$) in the vicinity of the upper warmer boundary ($z = 0.97$), corresponding to the slow heating mode. The different times are given starting from $\tau_1 = 0.00042$ [curve (1)] to $\tau_6 = \tau_{in} = 0.0134$ [curve (6)], respectively. (**b**) The same as in (**a**), but for the cooling mode at different times, from $\tau_7 = 0.014$ [curve (7)] to $\tau_{11} = \tau_R = 0.22$ [curve (11)], respectively.

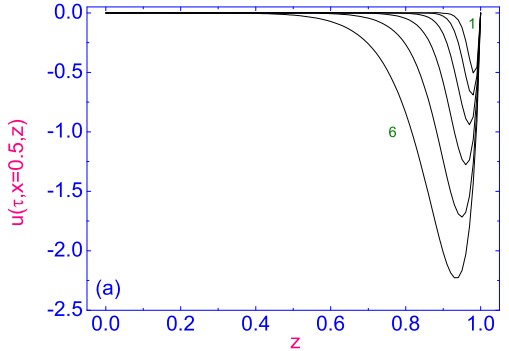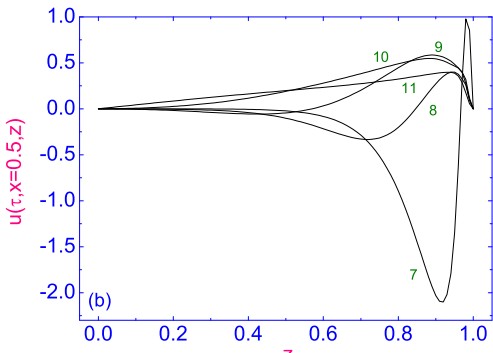

**Figure 4.** (**a**) The distance $z$ dependence of the horizontal component $u(\tau, x = 0.5, z)$ of the velocity **v** during its evolution to the equilibrium distribution $u_{eq}(x = 0.5, z) = u(\tau = \tau_6, x = 0.5, z)$ across the HAN channel [21]. The different times during the slow heating mode are given starting from $\tau_1 = 0.00042$ [curve (1)] to $\tau_6 = 0.0134$ [curve (6)], respectively. (**b**) The same as in (**a**), but for the cooling mode at different times, from $\tau_7 = 0.014$ [curve (7)] to $\tau_{11} = \tau_R = 0.22$ [curve (11)], respectively.

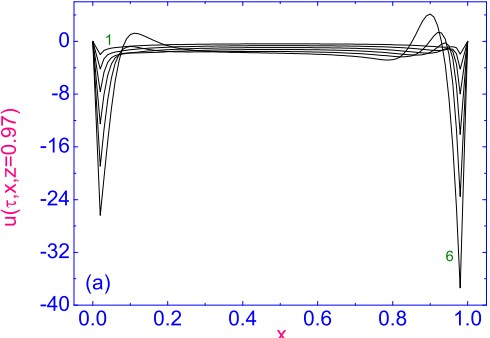 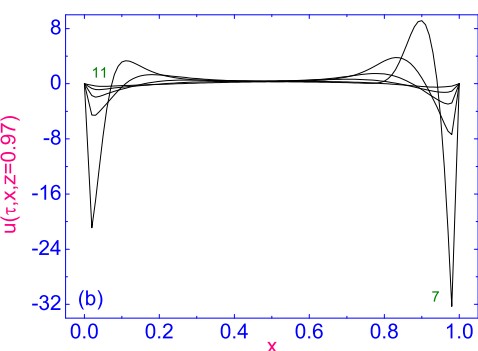

**Figure 5.** The same as in Figure 4a,b, but the evolution of the horizontal component $u(\tau, x, z = 0.97)$ of the velocity **v** to its equilibrium distribution $u_{eq}(x, z = 0.97) = u(\tau = \tau_6, x, z = 0.97)$ [21], along the width of the HAN channel ($0 \leq x \leq 1$) in the vicinity of the upper warmer boundary of the HAN channel ($z = 0.97$), during both the slow heating (**a**) and cooling (**b**) modes.

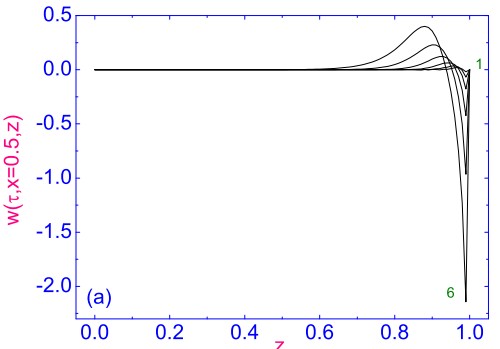 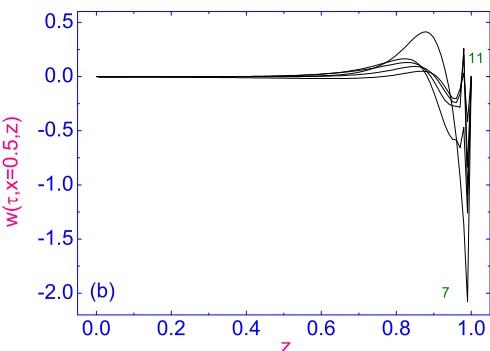

**Figure 6.** (**a**) The distance $z$ dependence of the vertical component $w(\tau, x = 0.5, z)$ of the velocity **v** during its evolution to the equilibrium distribution $w_{eq}(x = 0.5, z) = u(\tau = \tau_6, x = 0.5, z)$ across the HAN channel [21]. The different times during the slow heating mode are given starting from $\tau_1 = 0.00042$ [curve (1)] to $\tau_6 = 0.0134$ [curve (6)], respectively. (**b**) The same as in (**a**), but for the cooling mode at different times, from $\tau_7 = 0.014$ [curve (7)] to $\tau_{11} = \tau_R = 0.22$ [curve (11)], respectively.

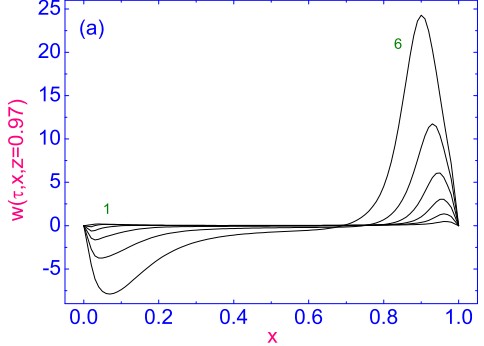 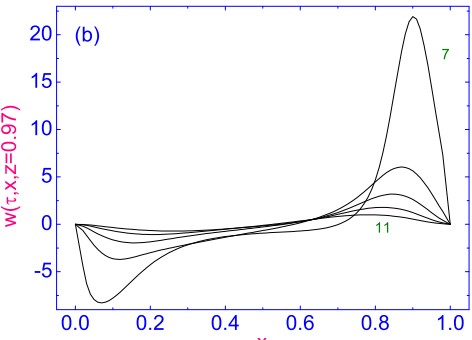

**Figure 7.** The same as in Figure 6a,b, but the evolution of the vertical component $w(\tau, x, z = 0.97)$ of the velocity **v** to its equilibrium distribution $w_{eq}(x, z = 0.97) = w(\tau = \tau_6, x, z = 0.97)$, along the width of the HAN channel ($0 \leq x \leq 1$) in the vicinity of the upper warmer boundary ($z = 0.97$), during both the slow heating (**a**) and cooling (**b**) modes [21].

The maximum of both the absolute magnitudes of the dimensional velocities $v_x(\tau, x, z) = (\frac{\gamma_1 d}{K_1})u(\tau, x, z)$ and $v_z(\tau, x, z) = (\frac{\gamma_1 d}{K_1})w(\tau, x, z)$ in the rectangular HAN channel, at the initial stage of the slow heating mode are equal to 1.2 mm/s and 0.88 mm/s, for the horizontal and vertical components (see Figure 5a and Figure 7a), at $\Delta \chi_{max} = 0.03$ ($\sim$9.3 $K$), respectively.

The distance $z$ dependence of the temperature field $\chi(\tau, x = 0.5, z)$ in the middle part of the dimensionless HAN channel ($x = 0.5$), during its evolution to the equilibrium distribution $\chi_{eq}(x = 0.5, z)$, under the effect of the dimensionless heat flux $Q = 0.44$ ($\sim$3.7 $\times$ $10^{-3}$ mW/$\mu$m$^2$), caused by the laser beam, at different times, from $\tau_1 = 0.00042$ [curve (1)] to $\tau_6 = \tau_{in} = 0.0134$ [curve (6)], respectively, is shown in Figure 8a.

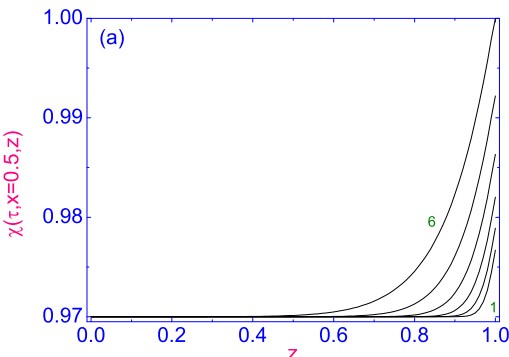 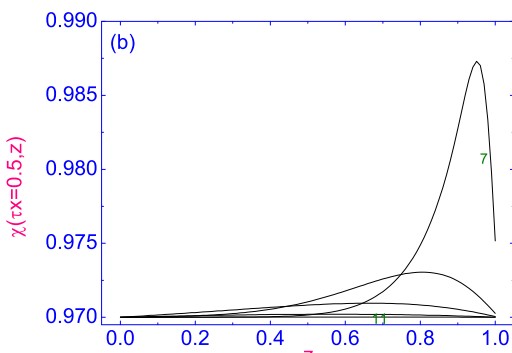

**Figure 8.** (**a**) The distance $z$ dependence of the temperature field $\chi(\tau, x = 0.5, z)$ during its evolution to the equilibrium distribution $\chi_{eq}(x = 0.5, z) = \chi(\tau = \tau_6, x = 0.5, z)$ in the middle part of the HAN channel [21], during the slow heating mode at different times, from $\tau_1 = 0.00042$ [curve (1)] to $\tau_6 = \tau_{in} = 0.0134$ [curve (6)], respectively. (**b**) The same as in (**a**), but for the cooling mode at different times, from $\tau_7 = 0.014$ [curve (7)] to $\tau_{11} = \tau_R = 0.22$ [curve (11)], respectively.

Figure 8b shows the same as in Figure 8a, but for the cooling mode at different times, from $\tau_7 = 0.014$ [curve (7)] to $\tau_{11} = \tau_R = 0.22$ [curve (11)], respectively. The distance $z$ dependence of the temperature field $\chi(\tau, x = 0.5, z)$ is characterized by the temperature growth on the upper boundary of the HAN channel ($z = 1$), from $\chi_{z=1} = 0.97$ ($\sim$298 $K$) to $\chi_{z=1} = 1.0$ ($\sim$307.3 $K$), during the slow heating mode (curves, from (1) to (6)) (see Figure 8a), with the following temperature decrease, from $\chi_{z=1} = 1.0$ ($\sim$307.3 $K$) to $\chi_{z=1} = 0.97$ ($\sim$298 $K$), after switching off the laser power (see Figure 8b). Here, the value of $\tau_{in} = \tau_6$ is equal to $\sim$0.0134 ($\sim$1.9 ms). The second part of the relaxation process of $\chi(\tau, x = 0.5, z)$ ($\tau \geq \tau_6$) is characterized by much faster cooling down of the upper restricted surface than the rest bulk of the nematic sample (see Figure 8b). The growth of the temperature field $\chi(\tau, x, z = 0.97)$ to its equilibrium distribution $\chi_{eq}(x, z = 0.97)$ along the width of the HAN channel ($0 \leq x \leq 1$) in the vicinity of the upper warmer boundary ($z = 0.97$), during the slow heating mode at different times, from $\tau_1 = 0.00042$ [curve (1)] to $\tau_6 = \tau_{in} = 0.0134$ [curve (6)], respectively, is shown in Figure 9a.

Figure 9b shows the same as in Figure 9a, but for the cooling mode at different times, from $\tau_7 = 0.014$ [curve (7)] to $\tau_{11} = \tau_R = 0.22$ [curve (11)], respectively. This slow heating mode is characterized by symmetric growth of the profile $\chi(\tau, x, z = 0.97)$ with respect to the middle part ($x = 0.5$) of the HAN channel. Such symmetric dependence of the temperature field along the width of the channel can be explained by much faster evolution of $\chi(\tau, x, z)$ to $\chi_{eq}(x, z)$, than the evolution both of the director and velocity fields to their equilibrium distributions across the HAN channel, and, as a result, the weak effect of these fields on $\chi$ [3].

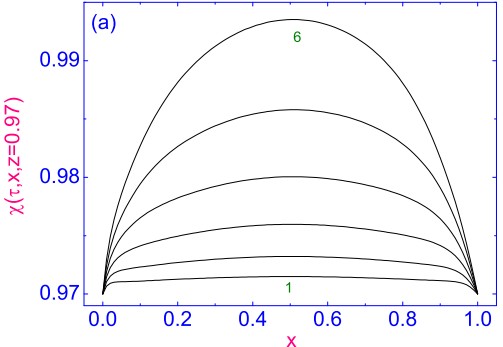
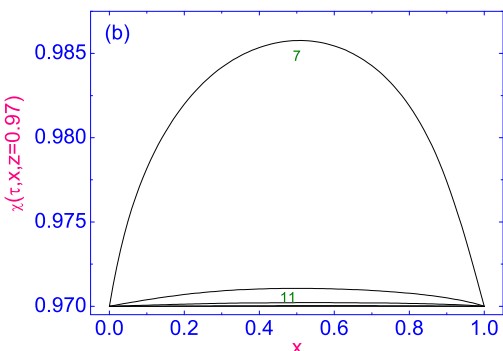

**Figure 9.** The same as in Figure 8a,b, but the distance $x$ dependence of the temperature field $\chi(\tau, x, z = 0.97)$ during its evolution to the equilibrium distribution $\chi_{eq}(x, z = 0.97) = \chi(\tau = \tau_6, x, z = 0.97)$ along the width of the HAN channel $(0 \leq x \leq 1)$, in the vicinity of the upper warmer restricted surface $(z = 0.97)$, both during the slow heating (**a**) and cooling (**b**) modes [21].

### B. Fast heating mode

The equilibrium distributions of the polar angle $\theta_{eq}(x, z) = \theta(\tau = \tau_5, x, z)$, both across the middle part $\theta_{eq}(x = 0.5, z) = \theta(\tau = \tau_5, x = 0.5, z)$ and in the vicinity of the upper boundary of the HAN channel $\theta_{eq}(x, z = 0.97) = \theta(\tau = \tau_5, x, z = 0.97)$, under the effect of the dimensionless heat flux $Q = 3.54$ ($\sim 2.95 \times 10^{-2}$ mW/$\mu$m$^2$) (which is 8 times greater than $Q$ in the first case A), caused by the laser beam focused on the upper boundary, are shown in Figure 10a,b, respectively.

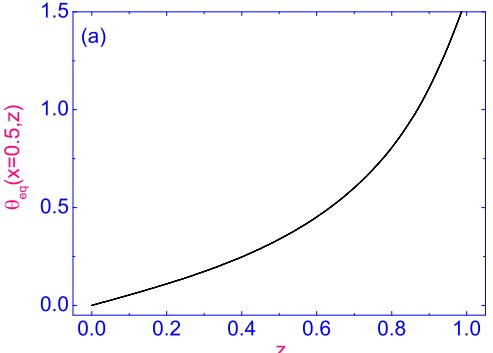
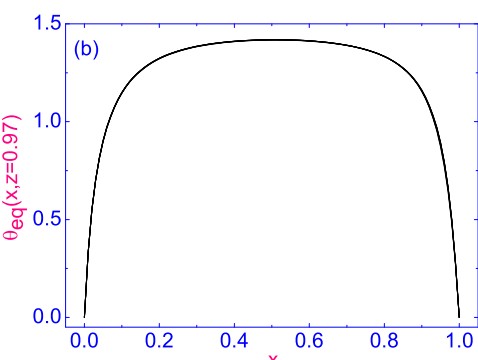

**Figure 10.** (**a**) The equilibrium distribution of the polar angle $\theta_{eq}(x = 0.5, z)$ in the middle part of the HAN channel, under the effect of the dimensionless heat flow $Q = 3.54$ ($\sim 2.95 \times 10^{-2}$ mW/$\mu$m$^2$) [21]. (**b**) The equilibrium distribution of the polar angle $\theta_{eq}(x, z = 0.97)$ along the width of the channel $(0 \leq x \leq 1)$ in the vicinity of the upper warmer boundary of the HAN channel $(z = 0.97)$, under the same conditions as in (**a**).

Here, the duration $\tau_{in}$ of the energy injection into the HAN channel across the upper boundary is equal to 0.0006 ($\sim 90$ $\mu$s), and during that time the temperature on the upper boundary grows from $\chi_{z=1}(\tau = 0) = 0.97$ (298 K) to $\chi_{z=1}(\tau = \tau_5 = \tau_{in} \sim 0.0006) = 1.0$ (307.3 K) (see Figure 10a), with the following cooling down to $\chi_{z=1}(\tau = \tau_R = 0.064) = 0.97$ (298 K). Note that the duration of the laser pulse, at fixed power $P_0 = 115$ mW, is limited by condition that the higher temperature on the upper boundary of the HAN channel $\chi_{z=1}$ must fall within the nematic stability range. The growth of the temperature field $\chi(\tau, x = 0.5, z)$ in the middle part of the dimensionless HAN channel $(x = 0.5)$ to its equilibrium distribution $\chi_{eq}(x = 0.5, z)$, under the effect of the dimensionless heat flux $Q = 3.54$ ($\sim 2.95 \times 10^{-2}$ mW/$\mu$m$^2$), caused by the laser beam at different times, from

$\tau_1 = 0.00004$ ($\sim$6 μs) [curve (1)] to $\tau_5 = 0.0006$ [curve (5)], respectively, corresponding to the fast heating mode, is shown in Figure 11a.

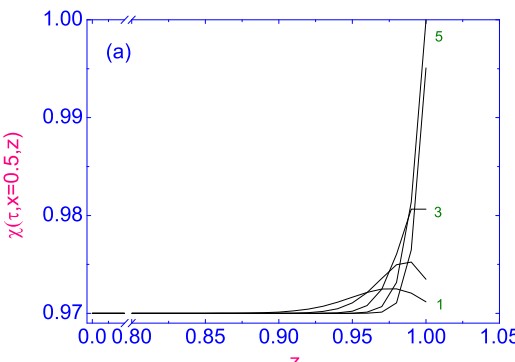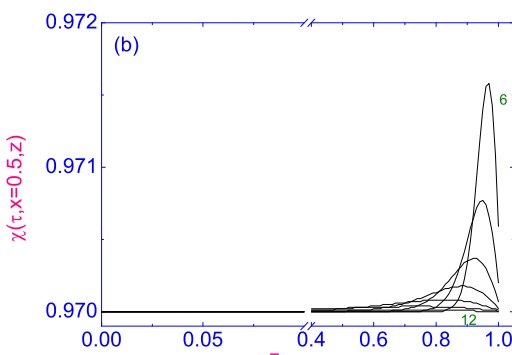

**Figure 11.** (**a**) The evolution of the temperature field $\chi(\tau, x = 0.5, z)$ to its equilibrium distribution $\chi_{eq}(x = 0.5, z) = \chi(\tau = \tau_5, x = 0.5, z)$ in the middle part of the HAN channel, under the effect of the dimensionless heat flux Q = 3.54 ($\sim$2.95 $\times$ 10$^{-2}$ mW/μm$^2$) [21]. The fast heating mode is characterized by the sequence of times, from $\tau_1 = 0.00004$ [curve (1)] to $\tau_5 = 0.0006$ [curve (5)], respectively, whereas the cooling mode (**b**) is characterized by the sequence of times, from $\tau_6 = 0.001$ [curve (6)] to $\tau_{12} = \tau_R = 0.064$ [curve (12)], respectively.

Here, $\tau_{in} = \tau_5 \sim 0.0006$ ($\sim$90 μs) is the duration of the energy injection into the HAN channel across the upper boundary by the infrared laser with the power $P_0 = 115$ mW. The sequence of times, from $\tau_6 = 0.001$ [curve (6)] to $\tau_{12} = \tau_R = 0.064$ [curve (12)], corresponding to the cooling mode, is shown in Figure 11b. The distance $x$ dependence of the temperature field $\chi(\tau, x, z = 0.97)$ during its evolution to the equilibrium distribution $\theta_{eq}(x, z = 0.97) = \theta(\tau = \tau_6, x, z = 0.97)$ [21], along the width of the HAN channel ($0 \leq x \leq 1$) in the vicinity of the upper warmer boundary ($z = 0.97$), during the fast heating mode at different times, from $\tau_1 = 0.00004$ [curve (1)] to $\tau_5 = 0.0006$ [curve (5)], respectively, is shown in Figure 12a, whereas the cooling mode with the sequence of times, from $\tau_6 = 0.001$ [curve (6)] to $\tau_{12} = \tau_R = 0.064$ [curve (12)], respectively, is shown in Figure 12b.

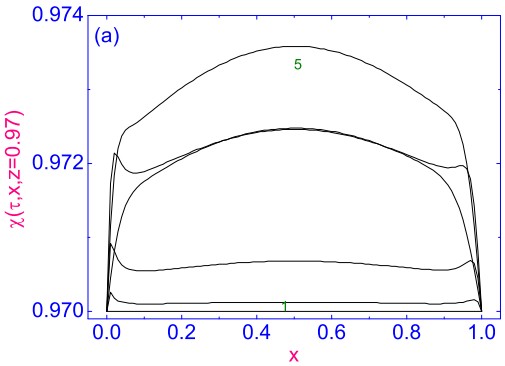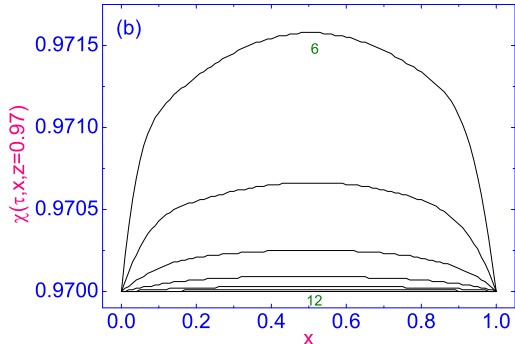

**Figure 12.** (**a**) The same as in Figure 11a, but the evolution of the temperature field $\chi(\tau, x, z = 0.97)$ to its equilibrium distribution $\chi_{eq}(x, z = 0.97) = \chi(\tau = \tau_5, x, z = 0.97)$ along the width of the HAN channel ($0 \leq x \leq 1$) in the vicinity of the upper warmer boundary ($z = 0.97$) [21] at different times, from $\tau_1 = 0.00004$ [curve (1)] to $\tau_5 = 0.0006$ [curve (5)], respectively, is shown for the fast heating mode. (**b**) The same as in (**a**), but the sequence of times, from $\tau_6 = 0.001$ [curve (6)] to $\tau_{12} = \tau_R = 0.064$ [curve (12)], corresponds to the cooling mode.

The distance $z$ dependence of the horizontal component $u(\tau, x = 0.5, z)$ of the velocity field in the middle part of the HAN channel ($x = 0.5$), during its evolution to

the equilibrium distribution $u_{eq}(x = 0.5, z) = u(\tau = \tau_5, x = 0.5, z)$, under the effect of the dimensionless heat flux $Q = 3.54$ ($\sim 2.95 \times 10^{-2}$ mW/μm²) at different times, from $\tau_1 = 0.00004$ ($\sim 6$ μs) [curve (1)] to $\tau_5 = 0.0006$ ($\sim 90$ μs) [curve (5)], respectively, corresponding to the fast heating mode is shown in Figure 13a.

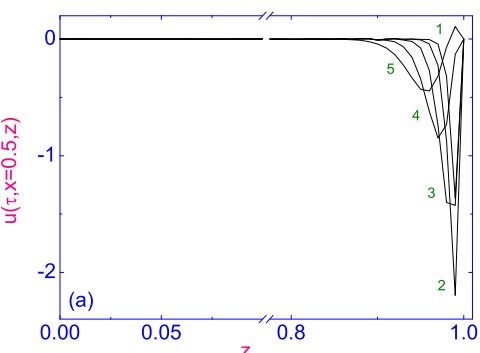 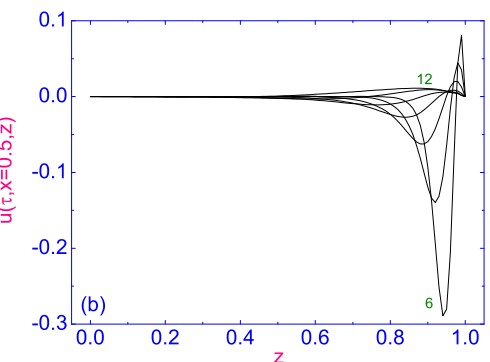

**Figure 13.** (**a**) The evolution of the horizontal component $u(\tau, x = 0.5, z)$ of the velocity **v** in the middle part of the HAN channel ($x = 0.5$) to the equilibrium distribution $u_{eq}(x = 0.5, z) = u(\tau = \tau_5, x = 0.5, z)$ across the HAN channel [21] at different times, from $\tau_1 = 0.00004$ [curve (1)] to $\tau_5 = 0.0006$ [curve (5)], respectively, corresponds to the fast heating mode. (**b**) The same as in (**a**), but the sequence of times, from $\tau_6 = 0.001$ [curve (6)] to $\tau_{12} = \tau_R = 0.064$ [curve (12)], respectively, is corresponding to the cooling mode.

Figure 13b shows the evolution of the horizontal component $u(\tau, x = 0.5, z)$ of the velocity **v** in the middle part of the HAN channel, during the cooling mode, with the sequence of times, from $\tau_6 = 0.001$ [curve (6)] to $\tau_{12} = \tau_R = 0.064$ [curve (12)], respectively. The distance $z$ dependence of $u(\tau, x = 0.5, z)$ (see Figure 13a) is characterized by strong increase of $|u(\tau, x = 0.5, z)|$, from $|u(\tau = 0)| = 0$ to $|u(\tau = \tau_2)| \sim 2.5$, within the first part of the fast heating mode ($\sim 11$ μs), in the vicinity of the upper warmer boundary, with following decrease up to $\sim 0$, during the rest time term $\tau_R - \tau_2 \sim 0.064$ ($\sim 96$ ms), respectively.

The distance $x$ dependence of the horizontal component $u(\tau, x, z = 0.97)$ of the velocity **v** during its evolution to the equilibrium distribution $u_{eq}(x, z = 0.97) = u(\tau = \tau_5, x, z = 0.97)$, along the width of the HAN channel ($0 \le x \le 1$) in the vicinity of the upper warmer restricted surface ($z = 0.97$) [21], for the fast heating mode at different times, from $\tau_1 = 0.00004$ [curve (1)] to $\tau_5 = 0.0006$ [curve (5)], respectively, is shown in Figure 14a.

Figure 14b shows the same as in Figure 14a, but the sequence of times, from $\tau_6 = 0.001$ [curve (6)] to $\tau_{12} = \tau_R = 0.064$ [curve (12)], respectively, corresponds to the cooling mode. Both the fast heating (Figure 14a) and cooling (Figure 14b) modes are characterized by near symmetric profile of $u(\tau, x, z = 0.97)$ with respect to the middle part ($x = 0.5$) of the HAN channel.

The distance $z$ dependence of the vertical component $w(\tau, x = 0.5, z)$ of the velocity **v**, during its evolution to the equilibrium distribution $w_{eq}(x = 0.5, z) = w(\tau = \tau_5, x = 0.5, z)$ across the HAN channel, corresponding to the fast heating mode at different times, from $\tau_1 = 0.00004$ [curve (1)] to $\tau_5 = 0.0006$ [curve (5)], respectively, also is characterized by oscillating behavior of $w(\tau, x = 0.5, z)$ in the vicinity of the upper warmer boundary (see Figure 15a), with the following velocity decreasing to 0, after switching off the laser power (see Figure 15b).

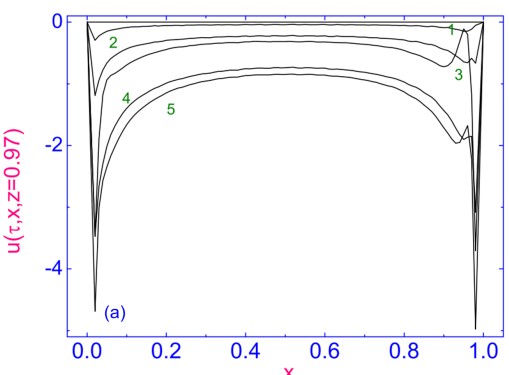
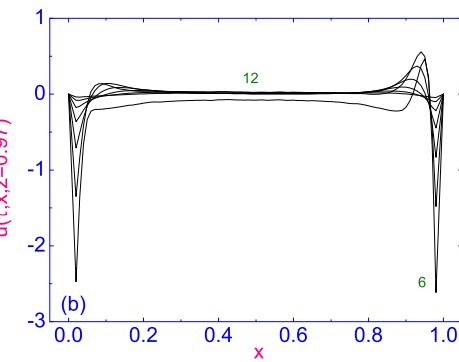

**Figure 14.** (**a**) The same as in Figure 13a, but the distance $x$ dependence of the horizontal component $u(\tau, x, z = 0.97)$ of the velocity **v** during its evolution to the equilibrium distribution $u_{eq}(x, z = 0.97) = u(\tau = \tau_5, x, z = 0.97)$, along the width of the HAN channel ($0 \leq x \leq 1$) in the vicinity of the upper warmer boundary ($z = 0.97$) [21], is shown for the fast heating mode at different times, from $\tau_1 = 0.00004$ [curve (1)] to $\tau_5 = 0.0006$ [curve (5)], respectively. (**b**) The same as in (**a**), but the sequence of times, from $\tau_6 = 0.001$ [curve (6)] to $\tau_{12} = \tau_R = 0.064$ [curve (12)], respectively, corresponds to the cooling mode.

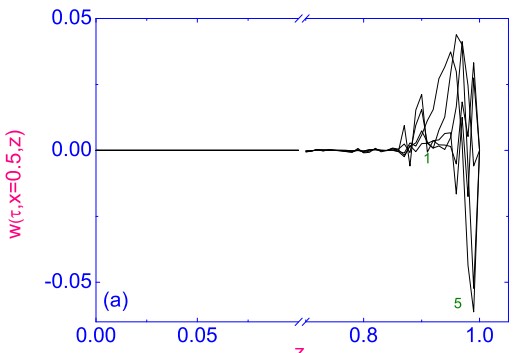
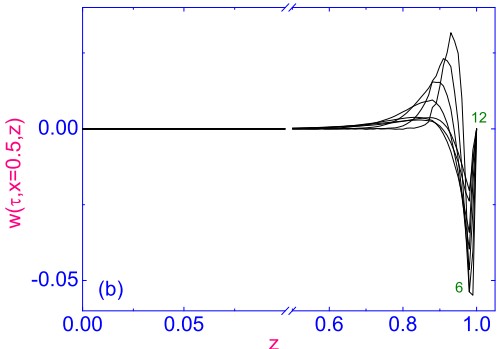

**Figure 15.** (**a**) The distance $z$ dependence of the vertical component $w(\tau, x = 0.5, z)$ of the velocity **v** during its evolution to the equilibrium distribution $w_{eq}(x = 0.5, z) = w(\tau = \tau_5, x = 0.5, z)$ across the HAN channel, corresponding to the fast heating mode [21], is given at different times, from $\tau_1 = 0.00004$ [curve (1)] to $\tau_5 = 0.0006$ [curve (5)], respectively. (**b**) The same as in (**a**), but the sequence of times, from $\tau_6 = 0.001$ [curve (6)] to $\tau_{12} = \tau_R = 0.064$ [curve (12)], corresponds to the cooling mode.

In turn, the distance $x$ dependence of the vertical component $w(\tau, x, z = 0.97)$ of the velocity **v** during its evolution to the equilibrium distribution $w_{eq}(x, z = 0.97) = w(\tau = \tau_5, x, z = 0.97)$, along the width of the HAN channel ($0 \leq x \leq 1$) in the vicinity of the upper warmer boundary ($z = 0.97$), both during the fast heating and cooling modes, are shown in Figure 16a,b, respectively.

The fast heating mode is characterized by exciting the vertical component $w(\tau, x, z = 0.97)$ of the velocity **v** only in the vicinity of the vertical boundaries of the HAN channel (see Figure 16a), with the following velocity decreasing to 0, after switching off the laser power (see Figure 16b), respectively. Note that under the effect of the heat flux across the upper boundary, only approximately 15% of the HAN channel (see Figure 15a) close to the upper warmer boundary is involved in the moving process, whereas the rest amount of the nematic sample is kept unmoved during the full relaxation term. These numerical studies show that in the hybrid aligned nematic material under the effect of **q**, in the vicinity of the warmer boundary, the vortical flow may be excited. The maximum for both the absolute magnitudes of the dimensional velocities $v_x(\tau, x, z) = \left(\frac{\gamma_1 d}{K_1}\right) u(\tau, x, z)$

and $v_z(\tau, x, z) = (\frac{\gamma_1 d}{K_1})w(\tau, x, z)$ in the HAN channel, at the initial stage of the fast heating mode, are equal to 0.2 mm/s and 0.05 mm/s, for horizontal and vertical components (see Figure 15a and Figure 16a), at $\Delta\chi_{max} = 0.03$ ($\sim$9.3 K), respectively.

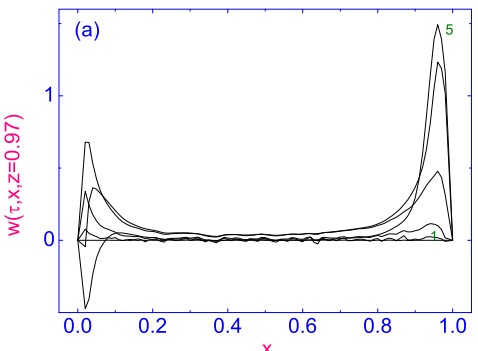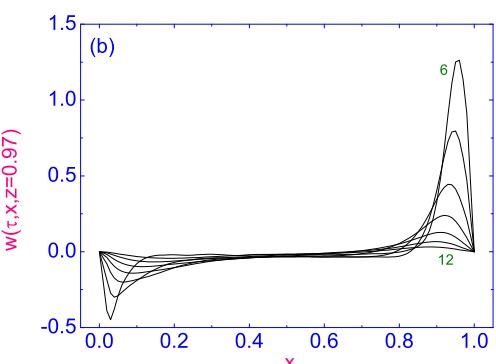

**Figure 16.** (**a**) The same as in Figure 15a, but the distance $x$ dependence of the vertical component $w(\tau, x, z = 0.97)$ of the velocity **v** during its evolution to the equilibrium distribution $w_{eq}(x, z = 0.97) = w(\tau = \tau_5, x, z = 0.97)$, along the width of the HAN channel ($0 \leq x \leq 1$) in the vicinity of the upper warmer boundary ($z = 0.97$), during the fast heating mode [21] is given at different times, from $\tau_1 = 0.00004$ [curve (1)] to $\tau_5 = 0.0006$ [curve (5)], respectively. (**b**) The same as in (**a**), but the sequence of times, from $\tau_6 = 0.001$ [curve (6)] to $\tau_{12} = \tau_R = 0.064$ [curve (12)], corresponds to the cooling mode.

In summary, we present the review of numerical results describing the evolution of not only the director $\hat{\mathbf{n}}(t, x, z)$ and velocity $\mathbf{v}(t, x, z)$ fields, but also the redistribution of the temperature field $\chi(t, x, z)$ in the 2$D$ hybrid aligned nematic channel to their equilibrium values, under the effect of the heat flux directed normal to the upper bounding surfaces, when the rest bounding surfaces of the HAN channel are kept at constant temperature. In this case, the upper nematic boundary is aligned homogeneously and heated by an infrared laser beam, and the dynamics of heating occurs with two distinct time scales:

(i)     a fast time scale, with the duration of the dimensional heat flux $q \sim 2.95 \times 10^{-2}$ mW/$\mu$m$^2$ over time $t_{in} \sim 90$ $\mu$s, and the laser power in $P_0 = 115$ mW, and

(ii)    a slow time scale, with the duration of the dimensional heat flux $q \sim 3.7 \times 10^{-3}$ mW/$\mu$m$^2$ over time $t_{in} \sim 1.9$ ms, and the laser power in $P_0 = 14.3$ mW, respectively [21].

In both these cases, the duration of the laser pulse, at fixed power in $P_0 = 14.3$ mW and $P_0 = 115$ mW, is limited by condition that the higher temperature on the upper bounding surface must fall within the nematic stability range. These numerical studies show that in such the hybrid aligned nematic volume under the effect of the heat flux, across the upper boundary, the vortical flow may be excited only in the vicinity of the warmer boundary. After some time more than the relaxation time, for instance, of the director field in the HAN channel, the vortical flow settles down to the rest state regime, where the horizontal $u$ and vertical $w$ components of velocity vector are equal to zero, and the temperature field across the nematic channel is finally falling to the value of temperature on the lower and both lateral bounding surfaces. Note that in the case of the fast heating mode (i) one deals with the heating which is characterized by "shallow" heating of the nematic microvolume in the vicinity of the upper bounding surface up to 40% of the full nematic sample, whereas in the case of the slow heating mode (ii), one deals with the "deeper" heating of the nematic microvolume up to 60% of the nematic sample far from the upper warmer restricted surface.

Thus, it can be concluded that for the formation of a hydrodynamic flow in a micro-sized HAN channel using laser radiation focused on the channel boundary, such a radiation power is necessary to pump energy into the liquid crystal phase for as long as possible. Another condition for the formation of a more powerful flow in the HAN channel is the planar anchoring condition at the boundary where the laser radiation will focus.

Therefore, a new scenario of the vortical flow formation in the HAN channel under the effect of the focused laser irradiation on the lower homeotropically aligned boundary will be analyzed in the next paragraph.

### 2.3. Laser Excited Vortical Flow in Microsized Nematic Channels

In the previous paragraph, we analyzed the process of formation and relaxation of the velocity field $\mathbf{v}$ in the microsized LC volume under the effect of the heat flux $\mathbf{q}$ directed orthogonally to the upper boundary of the HAN channel. The purpose of this paragraph is to show how the direction of the external heat flux $\mathbf{q}$ affects the process of formation of the vortex flow in the microsized HAN channel. In this case, the heat flux $\mathbf{q}$ directed across the lower bounding surface at an angle $\alpha$ with respect to the unit vector $\hat{\mathbf{i}}$. Here, $\hat{\mathbf{i}}$ is the unit vector parallel to the horizontal boundaries of the HAN channel, whereas the unit vector $\hat{\mathbf{k}}$ is directed perpendicular to both the horizontal boundaries. This problem will be treated in the framework of the extended Ericksen–Leslie theory [16,17], supplemented by the thermomechanical correction for both the shear stress and the Rayleigh dissipation function [7,15], as well as the entropy balance equation [18]. The hydrodynamic model in which the above problem will be considered is the same as in Section 2.1, and is based on the interaction effect of the director $\hat{\mathbf{n}}$ and temperature $T$ gradients with the velocity $\mathbf{v} = u\hat{\mathbf{i}} + w\hat{\mathbf{k}}$ field. In this case, the magnitude of the hydrodynamic flow $\mathbf{v}$ is proportional to $v \sim \frac{d}{\eta}\sigma_{zx}^{tm}$, where $\sigma_{zx}^{tm}$ is the tangential component of the thermomechanical stress tensor, $\eta$ is the viscosity of the nematic phase, and $d$ is the thickness of the HAN channel. It will be shown below that the tangential component of the thermomechanical stress, the tensor plays a crucial role in the formation of the thermally excited vortical flow in the HAN channel. In our case, the temperature gradient $\nabla T$ in the microsized nematic volume is formed by focused laser radiation. At the beginning of the section, we give a detailed description of the boundary conditions for both the director $\hat{\mathbf{n}}$ and temperature $T(x, z, t)$ fields, as well as for the velocity field $\mathbf{v}$. The second part of the section presents a numerical analysis of the formation of the thermally excited vortical flow in the HAN channel under the influence of the heat flux $\mathbf{q}$ directed across the lower bounding surface at an angle $\alpha$ with respect to the unit vector $\hat{\mathbf{i}}$.

With this aim, we consider the 2D nematic fluid composed of polar molecules, such as *cyanobiphenyls*, at density $\rho$, and delimited by two horizontal and two lateral surfaces at mutual distances $2D$ and $2d$, respectively. We also consider the effect of the orientational defect on the lower bounding surface on the vortex flow excited in the nematic microvolume under the effect of the focused laser beam. This defect is characterized by the continuous change in the orientation of the director's along the length of the lower restricted surface, from the homeotropic to the planar, and again to the homeotropic orientation.

Therefore, our 2D aligned nematic state with the orientational defect contains a gradient of $\nabla\theta$ on the lower boundary, i.e.,

$$\theta_{-10 \le x \le -L, z=-1} = \theta_{L \le x \le 10, z=-1} = 0,$$
$$\theta_{-L < x < L, z=-1} = \mathcal{A}, \tag{15}$$

and the planar orientation on the upper and lateral surfaces, i.e.,

$$\theta_{x=\pm 10, -1 < z < 1} = 0, \theta_{-10 \le x \le 10, z=1} = \frac{\pi}{2}, \tag{16}$$

where $\mathcal{A} = \tan^{-1}\left(\frac{L^2 - x^2}{4x^4}\right)$, $L = \frac{l}{d}$, and $2l$ is the length of the orientational defect on the lower boundary with the director's orientation changing continuously from the homeotropic to planar, and again to homeotropic orientation, whereas on the rest length of that surface there is the homeotropic director's orientation ($\hat{\mathbf{k}} \parallel \hat{\mathbf{n}}_{z=-1}$). Notice that in these calculations the ratio $D/d$ is equal to 10. Moreover, we will consider dimensionless spatial variables $\bar{x} = x/d$ and $\bar{z} = z/d$, and in the following equations the overbars will be eliminated.

In order to elucidate the role of both the orientational defect and the heat flux $\mathbf{q}$ directed at an angle $\alpha$ across the lower boundary of the HAN channel in formation of the vortical flow we consider the hydrodynamic regime with the heat flux $\mathbf{q} = q_z \hat{\mathbf{k}} = Q \sin \alpha \hat{\mathbf{k}}$, where $\alpha$ is the angle between vectors $\mathbf{q}$ and $\hat{\mathbf{i}}$. In the dimensionless form it can be written as [13,26]

$$(\chi_{,z}(\tau, x, z))_{z=-1} = \left[ \frac{q_z - (\lambda - 1)n_x n_z \chi_{,x}}{\lambda n_z^2 + n_x^2} \right]_{z=-1}, \tag{17}$$

whereas on the rest boundaries the temperature is kept constant

$$\chi_{-10 \leq x \leq 10, z=1} = \chi_{x \pm 10, -1 < z < 1} = \chi_0. \tag{18}$$

Here, $Q = Q(x, z = -1)$ is the injected energy across the lower boundary, $q_z = Q \sin \alpha$, and $\tau = \frac{t}{t_R}$ is the dimensionless characteristic time needed for reorientation of the director $\hat{\mathbf{n}}$, respectively.

The velocity $\mathbf{v} = u\hat{\mathbf{i}} + w\hat{\mathbf{k}} = -\nabla \times \hat{\mathbf{j}}\psi$ on these surfaces has to satisfy the no-slip boundary condition

$$u_{-10 \leq x \leq 10, z=-1} = (\psi_x)_{-10 \leq x \leq 10, z=-1} = u_{x=\pm 10, -1<z<1} = (\psi_x)_{x=\pm 10, -1<z<1} = 0,$$
$$w_{-10 \leq x \leq 10, z=-1} = (\psi_z)_{-10 \leq x \leq 10, z=-1} = w_{x \pm 10, -1<z<1} = (\psi_z)_{x \pm 10, -1<z<1} = 0, \tag{19}$$

where $u \equiv v_x(\tau, x, z)$ and $w \equiv v_z(\tau, x, z)$ are the components of the vector $\mathbf{v} = u\hat{\mathbf{i}} + w\hat{\mathbf{k}} = -\nabla \times \hat{\mathbf{j}}\psi$.

The reorientation of the director in the HAN microvolume confined between two horizontal and two vertical solid surfaces under the effect of the viscous, elastic, and thermomechanical forces and taking into account the flow can be obtained by solving the system of the nonlinear partial differential Equations (6), (8) and (14) with the appropriate dimensionless boundary conditions for the polar angle $\theta$ (Equations (15) and (16)), or for the director field $\hat{\mathbf{n}}$

$$(n_x)_{x=\pm 10, -1 \leq z \leq 1} = 0, \ (n_x)_{-10 < x < 10, z=1} = 1,$$
$$(n_x)_{-10 < x < -L, z=-1} = (n_x)_{L < x < 10, z=-1} = 0, (n_x)_{-L < x < L, z=-1} = \sin \mathcal{A}, \tag{20}$$

velocity field $\mathbf{v} = u\hat{\mathbf{i}} + w\hat{\mathbf{k}} = -\nabla \times \hat{\mathbf{j}}\psi$ (see Equation (19)), temperature field

$$\chi_{x=\pm 10, -1 \leq z \leq 1} = 0.97, \ \chi_{-10 < x < 10, z=1} = 0.97,$$
$$(\chi_{,z}(x, z))_{z=-1} = \left[ \frac{q_z - (\lambda - 1)n_x n_z \chi_{,x}}{\lambda n_z^2 + n_x^2} \right]_{z=-1}, \tag{21}$$

and the initial condition taken in the form

$$\hat{\mathbf{n}}(\tau = 0, x, z) = \hat{\mathbf{n}}_{elast}^{eq}(x, z), \tag{22}$$

respectively. Here, $\hat{\mathbf{n}}_{elast}^{eq}(x, z)$ is the equilibrium distribution of the director field over the nematic volume obtained from Equation (6), with $u_{,x} = u_{,z} = w_{,x} = w_{,z} = \chi_{,x} = \chi_{,z} = 0$, and with the boundary conditions in the form of Equation (20), whereas the initial condition is chosen in the form $\theta_{elast}(\tau = 0, x, z) = 0$, at $x = \pm 10$, and $-1 \leq z \leq 1$; $\frac{\pi}{4}(z + 1)$, at $-10 < x < -L$, and $L < x < 10$; $\frac{\pi}{2}$, at $-L \leq x \leq L$, and $z = -1$, respectively. Notice that the initial distribution of the director field $\hat{\mathbf{n}}_{elast}^{eq}(x, z)$ over the microsized volume has been obtained from Equation (6) by means of relaxation method [24]. In calculations, the relaxation criterion $\epsilon = |(\theta_{(m+1)}(\tau, x, z) - \theta_m(\tau, x, z))/\theta_{(m)}(\tau, x, z)|$ was chosen to be equal to $10^{-4}$, and the numerical procedure was then carried out until a prescribed accuracy was achieved. Here, $m$ is the iteration number.

Figure 17 shows a fragment of initial distribution of the director field $\hat{\mathbf{n}}^{eq}_{elast}(x, z)$ near the orientational defect located at $-L < x < L, z = -1$ [13,26]. Here, $L = \pm 0.5$ are the right and left ends of the orientational defect on the lower boundary. According to these calculations the highest value of $|\nabla \hat{\mathbf{n}}(x, z)|$ is reached in the vicinity of the orientational defect and its effect extends up to 3/4 of the thickness of the nematic volume.

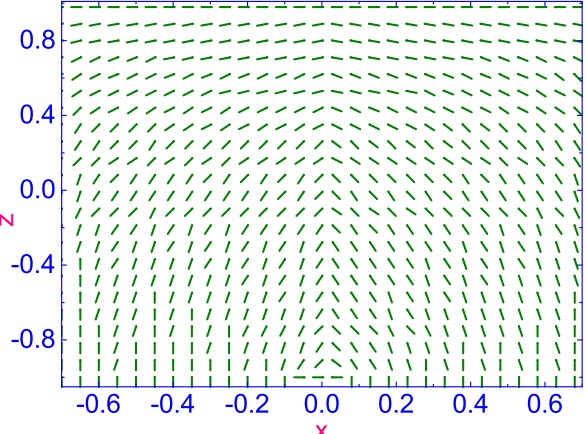

**Figure 17.** The fragment of initial distribution of the director field $\hat{\mathbf{n}}^{eq}_{elast}(x, z)$ near the orientational defect located at $-L < x < L, z = -1$. Here, $L = \pm 0.5$ [13,26].

The evolution of both the velocity $\mathbf{v} = u\hat{\mathbf{i}} + w\hat{\mathbf{k}} = -\nabla \times \hat{\mathbf{j}}\psi$ and the temperature $\chi(\tau = \tau_{in}, x, z)$ fields over the nematic microvolume with the orientational defect located on the lower boundary of the HAN channel as the function of the angle $\alpha$ are shown in Figures 18–23, respectively. The distribution of the velocity field $\mathbf{v}$ in the microscopic nematic volume with the orientational defect located at $-L \leq x \leq L, z = -1$, when the laser beam is directed at the values of the angle $\alpha$ equal to 20° and 40° (Figure 18a,b), 60° and 80° (Figure 19a,b), 90° (Figure 20), 100° and 120° (Figure 21a,b), 140° and 160° (Figure 22a,b) are shown in Figures 18–22 [13,26,27], respectively. The heating occurs during the dimensionless time $\tau_{in} = 1.6 \times 10^{-4}$ (~0.29 ms), whereas the value of dimensionless heat flux coefficient $Q_0 = \frac{d}{T_{NI}\lambda_\perp}q_0$ is equal to 0.05.

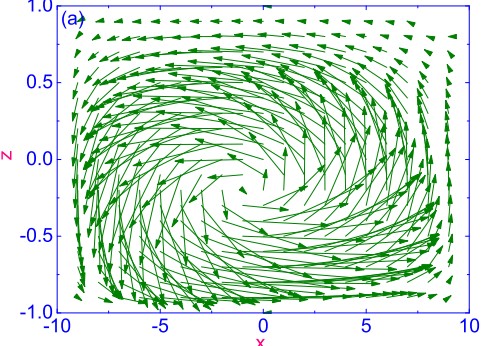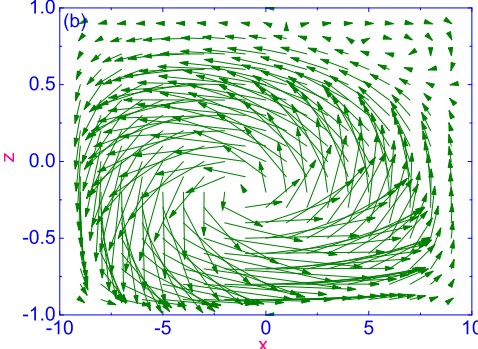

**Figure 18.** Distribution of the velocity field $\mathbf{v} = u\hat{\mathbf{i}} + w\hat{\mathbf{k}} = -\nabla \times \hat{\mathbf{j}}\psi$ in the microscopic HAN channel with the orientational defect located at $-L \leq x \leq L, z = -1$, when the heat flux $\mathbf{q}$ is directed at two values of the angle $\alpha$: (**a**) 20° and (**b**) 40° [12,26,27], respectively. The heating occurs during time $\tau_{in} = 1.6 \times 10^{-4}$ (~0.29 ms), whereas the value of dimensionless heat flux coefficient $Q_0$ is equal to 0.05. Here 1 mm of the arrow length is equal to 1.8 μm/s [13,26,27].

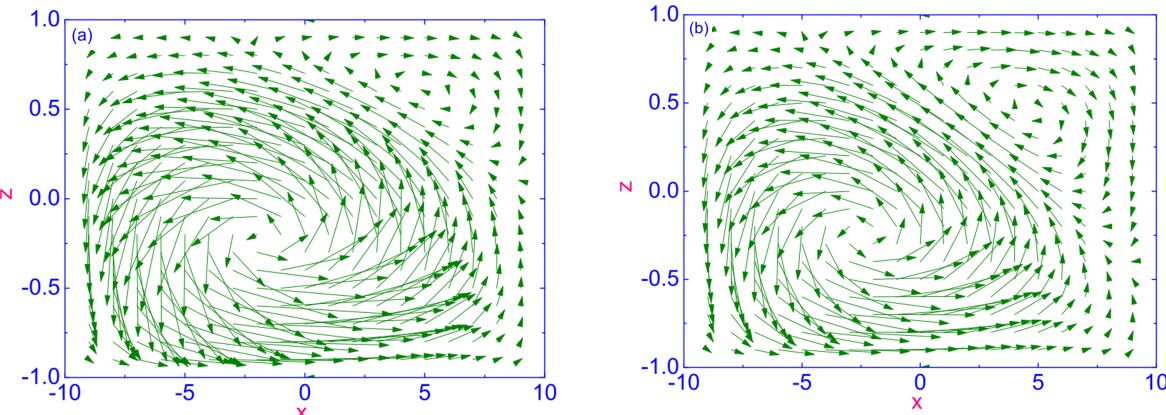

**Figure 19.** The same as in Figure 18, but the values of the angle $\alpha$ are: (**a**) 60° and (**b**) 80° [13,26,27], respectively.

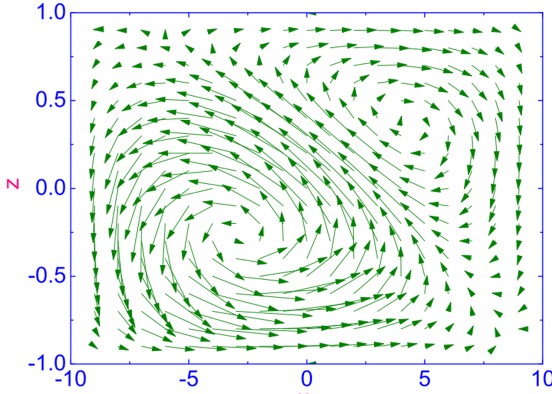

**Figure 20.** The same as in Figure 18, but the value of the angle $\alpha$ is equal to 90° [13,26,27].

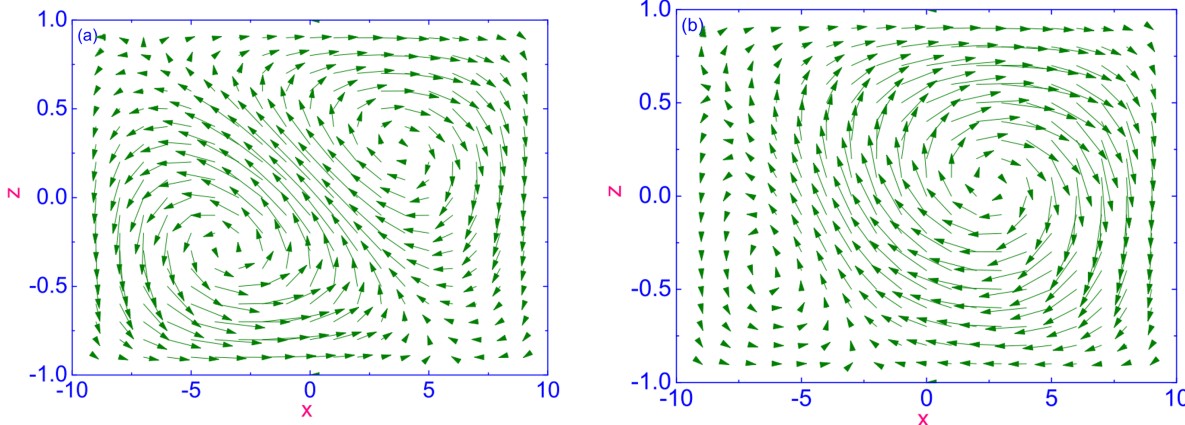

**Figure 21.** The same as in Figure 18, but the values of the angle $\alpha$ are: (**a**) 100° and (**b**) 120° [12,26,27], respectively.

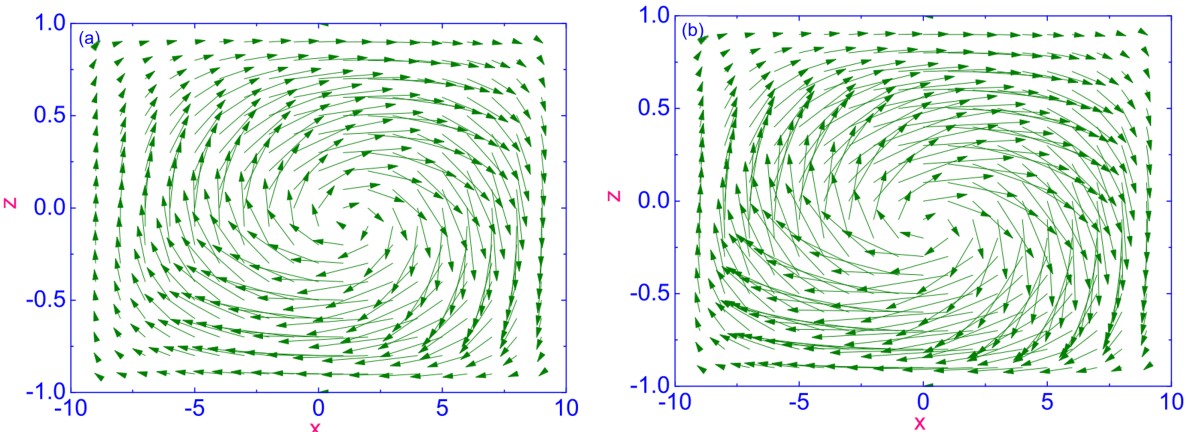

**Figure 22.** The same as in Figure 18, but the values of the angle $\alpha$ are: (**a**) 140° and (**b**) 160° [13,26,27], respectively.

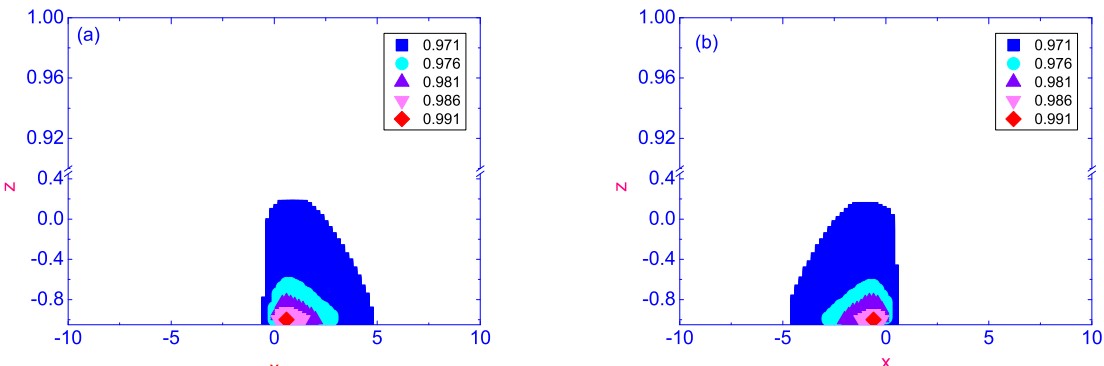

**Figure 23.** Temperature field distribution $\chi(\tau = \tau_{in}, x, z)$ over the HAN channel near the orientational defect located at $-L \le x \le L, z = -1$, when the heat flux **q** is directed at two values of the angle $\alpha$ [13]: (**a**) 20° and (**b**) 160°, respectively. The heating occurs during time $\tau_{in} = 1.6 \times 10^{-4}$ ($\sim$0.29 ms), whereas the value of dimensionless heat flux coefficient $Q_0$ is equal to 0.05.

The distribution of the velocity **v** in the microscopic HAN channel with the orientational defect located at $-L \le x \le L, z = -1$, when the heat flux **q** is directed at two values of the angle $\alpha$, 20° and 40° with respect to the lower bounding surface, is characterized by maintaining of vortices as shown in Figure 18a,b [13,26,27], respectively. Here, 1 mm of the arrow length is equal to 1.8 µm/s. The direction and magnitude of the hydrodynamic flow $\mathbf{v}(\tau, x, z)$ is influenced by both the direction of the heat flux **q** across the lower bounding surface and the character of the orientational defect. According to these calculations of the director field $\hat{\mathbf{n}}(\tau, x, z)$ across the nematic volume, the highest value of $|\nabla \hat{\mathbf{n}}(x, z)|$ is reached in the vicinity of the orientational defect, and, as a result, the biggest thermally excited velocities occur in the vicinity of the lower hotter surface. In that case, the self-sustaining vortical flow is excited in the negative sense (anti-clockwise) around its center, as the value of the angle $\alpha$ reaches 60°. Further increase in the angle value of more than 60 degrees leads to the formation of two vortices, one, larger, in the left-hand side of the nematic volume, another, smaller, in the right-side of the HAN channel, as shown in Figure 19a,b, Figure 20 and Figure 21a,b, respectively. With increasing the value of the angle $\alpha$, the larger vortex begins to decrease in size, while the smaller vortex begins to increase. The size of the smaller vortex reaches its maximum size at the angle $\alpha = 120°$ (see Figure 21b). Notice that the bigger self-sustaining vortical flow in the left-hand side of the HAN channel is thermally excited in the negative sense (anti-clockwise), whereas the smaller vortical flow in the right-hand side of the nematic volume is excited in the positive sense (clockwise) around their centers. With increasing the value of the angle $\alpha$ to more than 120 degrees, the



larger vortex disappears and we have only one self-sustaining vortical flow in the positive sense (clockwise) around its center (see Figure 22a,b).

These results show that the direction of the heat flux across the hotter bounding surface with the orientational defect plays a crucial role in maintaining the thermally excited vortical flow in 2D HAN channel.

The distribution of the temperature field $\chi(\tau = \tau_{in}, x, z)$ over the HAN microvolume near the orientational defect located at $-L \leq x \leq L, z = -1$, when the dimensional heat flux $\mathbf{q} = -\frac{T_{NI}\lambda_\perp}{d}\mathbf{Q}$ is directed at two values of the angle $\alpha$: $20°$ (Figure 23a) and $160°$ (Figure 23b), after time term $\tau_{in} = 1.6 \times 10^{-4}$ ($\sim$0.29 ms), is shown in Figure 23 [13].

The value of dimensionless heat flux coefficient $q_0 = \frac{T_{NI}\lambda_\perp}{d}Q_0$ is equal to 0.05 ($\sim$4.2 $\times$ $10^{-4}$ mW/$\mu$m$^2$). In general, under the above conditions, the picture of warming is such that only a small part of the nematic volume is involved in the heating process, while a large part of the volume of the fluid were not heated. It should be noted that in both cases the area of the greatest heating was shifted in the direction in which the heat flux was directed due to laser radiation.

We have reviewed the orientational dynamics in a microsized hybrid aligned nematic channel, where the nematic volume is confined by two horizontal and two lateral surfaces, and under the influence of the temperature gradient $\nabla T$, when the nematic material is heated by a laser beam focused on the lower boundary with the orientational defect, whereas the rest bounding surfaces of the LC volume are kept at constant temperature. It has been shown that in the microsized HAN channel, due to interaction between $\nabla T$ and the gradient of the director field $\nabla\hat{\mathbf{n}}$, in the LC volume with the orientational defect on the lower hotter boundary the vortical flow can be excited. The direction and magnitude of the hydrodynamic flow is influenced both by the heat flux $\mathbf{q}$ across the lower hotter boundary of the HAN channel, directed at the angle $\alpha$ with respect to the unit vector $\hat{\mathbf{i}}$, and by the character of the orientational defect. Calculations show that the biggest thermally exited vortical flow occurs in the vicinity of the orientational defect. The analysis showed that at the same power of the laser radiation, with a change in the value of the angle $\alpha$, the picture of the vortex flow in the HAN channel changes. When the heat flux $\mathbf{q}$ is directed at the angle $\alpha$ less than 40 degrees, in the nematic volume one the self-sustaining vortical flow in the negative sense (anti-clockwise) is excited around its center.

The increase in the angle value to more than 60 degrees leads to the formation of two vortices, one, larger, in the left-hand side of the LC volume, another, smaller, in the right-hand side of the LC volume, respectively. With further increasing the value of the angle $\alpha$ up to right angle and more, the larger vortex begins to decrease in size, while the smaller vortex reaches its maximum size at the angle 120 degrees. When the value of the angle $\alpha$ is more than 120 degrees, a single self-sustaining vortex flow is formed in a positive sense (clockwise) around its center [2].

Based on the analysis of the nature of the thermally excited vortical flow in the microsized HAN volume under the influence of the heat flux $\mathbf{q}$ directed across the boundary, we can conclude that the formation of the hydrodynamic flow requires such the radiation power that would pump energy into the liquid crystal phase for as long as possible. Another condition for the formation of the more powerful flow in the HAN channel is a planar anchoring condition on the boundary on which the laser radiation would be focused.

It should be noted that the vortical flow in the bulk of the microsized nematic volume is a unique phenomena only exhibited by liquid crystal systems, and expected to be applied for novel opto-thermal tweezers.

Now, in the next paragraph, we turn to the description of the formation of vortex flows in microsized HAN channels with a free upper surface under the effect of focused laser irradiation in the bulk of the nematic phase.

### 2.4. Laser Excited Motion of Nematics Confined in Microsized Channel with a Free Surface

Despite the fact that certain qualitative and quantitative advances in a hydrodynamic description of the relaxation processes in the microsized nematic volume under the effect

of the temperature gradient have been achieved, it is still too early to talk about the development of a theory which would make it possible to describe the dissipation processes in confined nematic channel with a free upper LC/air interface under the influence of the temperature gradient $\nabla$T [11,12]. Thus, we are primarily concerned here on describing the set of numerical results which show the way how the temperature gradient caused by the laser radiation focused in the interior of the microsized hybrid aligned nematic (HAN) volume with a free upper LC/air interface can produce the hydrodynamic flow and, as a result, how it can deform the free LC/air interface [11,12].

This problem will be treated in the framework of the appropriate nonlinear extension of the Ericksen–Leslie theory [16,17], supplemented by the thermomechanical correction of shear stress [7,15], and the entropy balance equation [18]. The hydrodynamic model in which the above problem will be considered is the same as in Section 2.1, and is based on the interaction effect of the director $\hat{\mathbf{n}}$ and temperature $T$ gradients with the velocity $\mathbf{v} = u\hat{\mathbf{i}} + w\hat{\mathbf{k}}$ field. The magnitude of the hydrodynamic flow $\mathbf{v}$ is proportional to $v \sim \frac{d}{\eta}\sigma_{zx}^{tm}$, where $\sigma_{zx}^{tm}$ is the tangential component of the thermomechanical stress tensor, $\eta$ is the viscosity of the nematic phase, and $d$ is the thickness of the HAN channel. It will be shown below that the tangential component of the thermomechanical stress tensor plays a crucial role in the formation of the thermally excited vortical flow in the HAN channel with the free upper surface. In this case, the temperature gradient $\nabla T$ in the microsized HAN volume is formed by a laser beam focused in the interior of the HAN channel. At the beginning of the section, we give a detailed description of the boundary conditions for both the director $\hat{\mathbf{n}}$ and temperature $T(x, z, t)$ fields, as well as for the velocity field $\mathbf{v}$. The bounding condition for the velocity on the upper free LC/air interface $\Gamma$ can be obtained from the linear momentum balance equation transmitted to the surface $\Gamma$. The temperature regime without the heat flux $\vec{q}$ across the free LC/air interface $\Gamma$ will be analyzed. The second part of the section presents a numerical analysis of two modes of the thermally excited vortical flow in the HAN channel with the free upper surface. These modes are characterized by different laser radiation power and duration: slow and fast heating modes caused by the laser irradiation focused in the interior of the HAN channel with the free upper surface.

With this aim, we consider the HAN channel delimited by one lower horizontal solid surface, located at $z = -d$, one upper free flat LC/air interface, initially located at $z = d$, and two lateral solid surfaces at distance $2L$ on scale on the order of micrometers. The coordinate system defined by our task is the same as in the previous paragraph.

Therefore, the hybrid aligned nematic phase contains a gradient of $\nabla\hat{\mathbf{n}}$ from planar orientation on the lower and both lateral surfaces to homeotropic orientation on the upper free LC/air interface $\Gamma$, i.e.,

$$(n_x)_{x=\pm L, -d<z<d} = 0, \quad (n_x)_{-L<x<L, z=-d} = 1,$$
$$(\vec{n} \cdot \vec{v})_{\Gamma} = -1. \tag{23}$$

Here, $\vec{v} = \left[-\frac{H_{,x}}{\sqrt{H_{,x}^2+1}}, 1\right]$ is the normal to the free LC/air interface $\Gamma$ at any time and is directed from the nematic phase into air, $H(x, t)$ is the height of the LC film on the top of the smooth surface, and $H_{,x} = \frac{\partial H}{\partial x}$. We consider the temperature regime without the heat flux $\vec{q}$ across the free LC/air interface $\Gamma$

$$(\vec{q} \cdot \vec{v})_{\Gamma} = 0, \tag{24}$$

whereas on the rest boundaries the temperature is kept constant

$$T_{-L<x<L, z=-d} = T_{x=\pm L, -d<z<d} = T_0. \tag{25}$$

We will assume the no-slip boundary conditions for the nematogenic molecules on these solid bounding surfaces, i.e.,

$$\mathbf{v}_{-L<x<L, z=-d} = \mathbf{v}_{x=\pm L, -d<z<d} = 0, \tag{26}$$

where $\mathbf{v} = u\hat{\mathbf{i}} + w\hat{\mathbf{k}} = -\nabla \times \hat{\mathbf{j}}\psi$ is the velocity vector with the horizontal $u \equiv v_x(t,x,z)$ and vertical $w \equiv v_z(t,x,z)$ components. The bounding condition for the velocity on the upper free LC/air interface $\Gamma$ can be obtained from the linear momentum balance equation transmitted to the surface $\Gamma$. In our case, that balance leads to the tangential

$$\left[\vec{v} \cdot \sigma \cdot \vec{t}\right]_\Gamma = 0, \tag{27}$$

and normal

$$\left[\vec{v} \cdot \sigma \cdot \vec{v}\right]_\Gamma = 2\gamma\kappa, \tag{28}$$

force balances, where $\vec{t} = \left[1, \frac{H_{,x}}{\sqrt{H_{,x}^2+1}}\right]$ is an additional unit tangent vector, $\gamma$ is the LC/air surface tension, $\kappa = \frac{H_{,xx}}{\sqrt{1+H_{,x}^2}}$ is the curvature of free LC/air interface $\Gamma$ at any time, and $\sigma$ is the full stress tensor (ST). Taking into account the microsized HAN volume, one can assume the mass density $\rho$ to be constant across the sample, and thus one deal with an incompressible fluid. The incompressibility condition $\nabla \cdot \mathbf{v} = 0$ assumes that

$$u_{,x} + w_{,z} = 0. \tag{29}$$

The hydrodynamic equations describing the reorientation of the nematic phase in 2D case, when the system is subjected to a temperature gradient $\nabla T$, due to uniform heat flux $\mathbf{q}$, can be derived from the torque balance equation Equation (6), the linear momentum equation for the velocity field $\mathbf{v}$, which can be written in the form of Equation (14), and the heat conduction equation [11,12]

$$\rho C_P \frac{dT}{dt} = -\nabla \cdot \mathbf{q} + \mathcal{O}(t,x,z), \tag{30}$$

where $\mathbf{q} = -T\frac{\delta \mathcal{R}}{\delta \nabla T}$ denotes the dimensional heat flux in the HAN channel, $\mathcal{O}(t,x,z) = \mathcal{O}_0 \exp\left[-2\frac{(x-x_0)^2+(z-z_0)^2}{\Delta^2}\right]\mathcal{H}(t_{in}-t)$ is the dimensional heat source, $\mathcal{H}(t_{in}-t)$ is the Heaviside step function, $\mathcal{O}_0 = \frac{2}{\pi}\frac{\alpha P_0}{\Delta^2}$ is the dimensional heat source coefficient, $\alpha$ is the coefficient of absorption, $P_0$ is the laser beam power, $\Delta$ is the Gaussian spot size, and $t_{in}$ is the duration of the energy injection into the HAN channel.

Now the dynamics of the height $H(t,x)$ of the LC/air interface under the influence of the temperature gradient can be obtained by solving the system of nonlinear partial differential Equations (6), (14) and (30) with the appropriate boundary and initial conditions. Equations (27) and (28), together with the torque balance Equation (6), transmitted to the LC/air interface, can be combined to yield equation for the height $H(t,x)$ in the form [11,12]

$$\frac{\partial H}{\partial t} = w_\Gamma - u_\Gamma H_{,x}, \tag{31}$$

where $u_\Gamma$ and $w_\Gamma$ are the horizontal and vertical components of the velocity $\mathbf{v}$ on the LC/air interface $\Gamma$, respectively.

Below, the set of dimensionless analog balance Equations (6), (14) and (30) will be considered. The dimensionless entropy balance equation now can be written as [11,12]

$$\chi_{,\tau} = \left[\chi_{,x}\left(\lambda n_x^2 + n_z^2\right) + (\lambda - 1)n_x n_z \chi_{,z}\right]_{,x} +$$
$$\left[\chi_{,z}\left(\lambda n_z^2 + n_x^2\right) + (\lambda - 1)n_x n_z \chi_{,x}\right]_{,z} +$$
$$\delta_4 \chi \left(\nabla \cdot \frac{\partial \mathcal{R}^{tm}}{\partial \nabla \chi}\right) + \delta_5 \mathcal{O}(\tau, x, z) - \psi_{,z}\chi_{,x} + \psi_{,x}\chi_{,z}, \tag{32}$$

where the extra one parameter of the LC system is $\delta_5 = \frac{2\alpha}{\pi\omega^2}\frac{d^2}{\lambda_\perp T_{NI}}\mathcal{O}_0$. Taking into account that the dimensionless temperature $\chi$ should be in the range of $[0.97 - 1.0]$, the parameter $\delta_5$ can be estimated as $\delta_5 \sim 7.0$. This estimation of $\delta_5 = \frac{2\alpha}{\pi\omega^2}\frac{d^2}{\lambda_\perp T_{NI}}\mathcal{O}_0$ was made taking into account the fact that the value of the dimensional heat flux coefficient is equal to $\mathcal{O}_0 \sim 0.5\,W$, whereas the heating occurs during time $t_{in} \sim 2.0\,\mu s$ (slow heating regime).

The evolution of the free LC/air interface under the effect of the temperature gradient $\nabla\chi$, caused by the laser irradiation focused in the interior of the HAN channel, is governed by Equations (6), (14) and (32), together with the boundary conditions at the solid boundaries [11,12]

$$(n_x)_{x=\pm 10, -1 \leq z \leq 1} = 0, \ (n_x)_{-10 \leq x \leq 10, z=-1} = 1,$$
$$\chi_{x=\pm 10, -1 \leq z \leq 1} = 0.97, \ \chi_{-10 \leq x \leq 10, z=-1} = 0.97,$$
$$(\psi_{,x})_{x=\pm 10, -1 \leq z \leq 1} = (\psi_{,z})_{x=\pm 10, -1 \leq z \leq 1} = 0,$$
$$(\psi_{,x})_{-10 \leq x \leq 10, z=-1} = (\psi_{,z})_{-10 \leq x \leq 10, z=-1} = 0, \tag{33}$$

and at the flexible free LC/air interface $\Gamma$ [1]

$$(\vec{n} \cdot \nabla\chi)_\Gamma = 0,$$
$$(\vec{n} \cdot \vec{v})_\Gamma = -1,$$
$$\hat{\mathcal{B}} \cdot \vec{\Psi} = \vec{\mathcal{C}}, \tag{34}$$

respectively, and the initial condition in the form of Equation (22). Here, $\vec{\Psi} = (\psi_{,xx}, \psi_{,xz}, \psi_{,zz})$, and both the matrix $\hat{\mathcal{B}}$ and vector $\vec{\mathcal{C}}$ are given in the Appendix, whereas the vector $\hat{\mathbf{n}}_{el}(x, z)$ is obtained from Equation (6), with $\psi_{,x} = \psi_{,z} = \chi_{,x} = \chi_{,z} = 0$.

Now the dimensionless height $\overline{H}(\tau, x) = H(\tau, x)/d$ of the nematic–air interface at any time $\tau$ can be calculated as [11,12]

$$H_{,\tau} + (\psi_{,x})_\Gamma + (\psi_{,z})_\Gamma H_{,x} = 0, \tag{35}$$

where $w(\tau, x)_{x \in \Gamma} = (\psi_{,z})_{x \in \Gamma}$ and $u(\tau, x)_{x \in \Gamma} = (\psi_{,x})_{x \in \Gamma}$ are the vertical and horizontal components of the velocity vector $\mathbf{v} = u\hat{\mathbf{i}} + w\hat{\mathbf{k}} = -\nabla \times \hat{\mathbf{j}}\psi$ on the interface $\Gamma$. Notice that the overbar in the function $H$ has been (and will be) eliminated in the last, as well as in the following equations.

Thus, when the director $\hat{\mathbf{n}}$ is strongly homogeneously anchored to the lower boundary and planar to the lateral boundaries of the HAN channel, the value of $\hat{\mathbf{n}}$ has to satisfy the boundary conditions (33) and (34) and its initial orientation (22), and then, under the action of the viscous, elastic, and thermomechanical forces, is allowed to relax to its equilibrium value $\hat{\mathbf{n}}_{el}(x, z)$.

A. Slow heating mode when a laser beam is focused in the interior of the HAN channel

Let us initially consider the case when the hybrid aligned nematic microvolume is heated by the laser beam focused in the interior of the HAN channel with the free upper boundary. In this case the value of the dimensional heat flux coefficient is equal to $\mathcal{O}_0 \sim 0.5\,W$, whereas the heating occurs during time $t_{in} \sim 2.0\,\mu s$.

Figure 24 shows the distance $x$ dependence of both the dimensionless height $h(x, \tau) = H(x, \tau) - 1$ (Figure 24a) of the LC/air interface and the dimensionless temperature $\chi(\tau, x)$ (Figure 24b) on the free LC/air interface $\Gamma$ [11,12], during the slow heating mode when the laser beam is focused in the interior ($x = 0.0$ and $z = 0.93$) of the HAN channel, at different times $\tau_i = 2^i \times 10^{-5}$ ($i = 1, \ldots, 10$), respectively, whereas Figure 25 shows the distance $x$ dependence of both the horizontal $u(\tau, x)$ (Figure 25a) and vertical $w(\tau, x)$ (Figure 25b) components of the vector $\mathbf{v} = u\hat{\mathbf{i}} + w\hat{\mathbf{k}} = -\nabla \times \hat{\mathbf{j}}\psi$ on the LC/air interface. According to these calculations [10,11], the evolution of the height $h(\tau, x)$ of the LC/air interface is characterized by the wavelike profile along the $x$-axis ($-0.5 < x < 0.5$). At the final stage of the evolution process, for $\tau = \tau_{in}$, the highest value of $|h| \sim 2 \times 10^{-2}$ is reached in the vicinity of points $x \sim \pm 0.25$, whereas the evolution of the temperature $\chi$ is characterized by symmetric profile of $\chi(\tau, x)_{x \in \Gamma}$ with respect to the middle point ($x = 0.0$) of the LC/air interface $\Gamma$ (see Figure 24b). In that case, during the heat step ($\tau \sim \tau_{in} \sim 0.01$ ($\sim 2$ μs)) the evolution of the temperature profile $\chi(\tau, x)_{x \in \Gamma}$ is characterized by its strong growth in the vicinity of the middle point $x = 0.0$, up to the highest value of 0.987 ($\sim 307$ K), whereas the evolution of the dimensionless height $h$ of the LC/air interface is characterized by two combs with the highest value of $|h| \sim 0.02$ ($\sim 0.01$ μm), which are directed in the opposite sense with respect to their center $x = 0.0$ (see Figure 24a). The thermally excited flow in the case of the slow heating mode, with $\mathcal{O}_0 = 0.5$ W and $t_{in} \sim 2.0$ μs, is characterized by maintaining of two vortices as shown in Figures 25a,b and 26 [11,12].

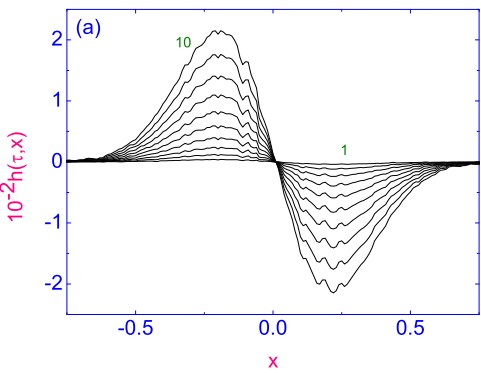 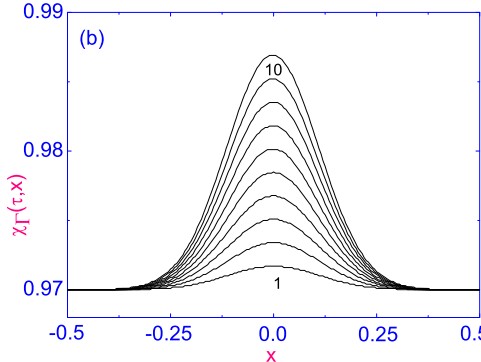

**Figure 24.** (**a**) The distance $x$ dependence of both the dimensionless height $h(\tau, x)$ of the LC/air interface and the dimensionless temperature $\chi(\tau, x)$ (**b**) on the free LC/air interface $\Gamma$, during the slow heating mode with $\delta_5 = 7$ and $\tau_{in} = 0.01$, at different times $\tau_i = 2^i \times 10^{-5}$ ($i = 1, \ldots, 10$) [11,12], respectively. The numbering of the curves increases from $i = 1$ to $i = 10$.

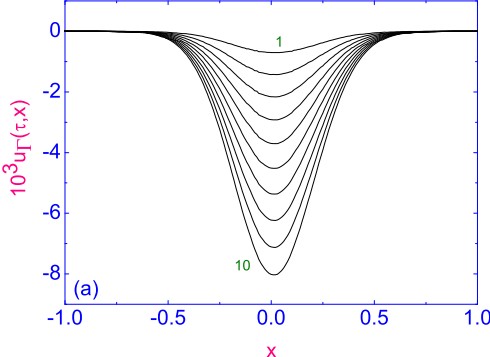 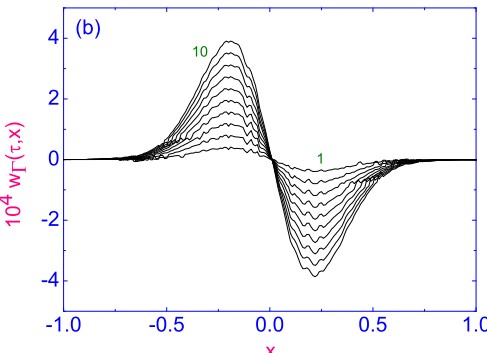

**Figure 25.** The same as in Figure 24, but the distance $x$ dependence of both the horizontal $u(\tau, x)$ (**a**) and vertical $w(\tau, x)$ (**b**) components of the velocity vector $\mathbf{v} = u\hat{\mathbf{i}} + w\hat{\mathbf{k}} = -\nabla \times \hat{\mathbf{j}}\psi$ on the LC/air interface during the slow heating mode [11,12].

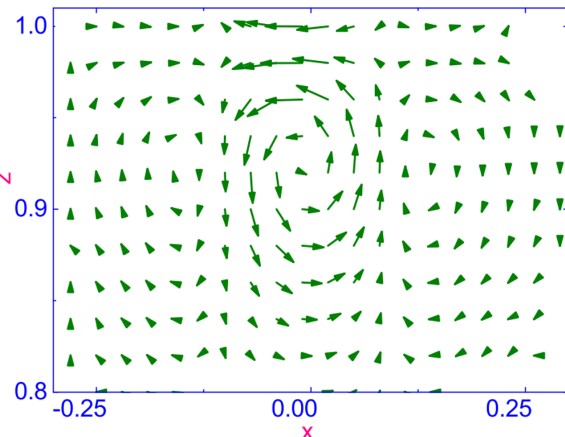

**Figure 26.** (Slow heating mode). Distribution of the velocity field $\mathbf{v} = u\hat{\mathbf{i}} + w\hat{\mathbf{k}} = -\nabla \times \hat{\mathbf{j}}\psi$ in the HAN channel after the slow heating mode during $\tau = \tau_{in}$ [11,12]. Here, 1 mm of the arrow length is equal to 0.04 μm/s.

According to these calculations, the thermally driven bi-vortical flow is maintained in the left-hand side of the HAN channel, whereas the focuses of both vortices are shifted to the left-hand side of the nematic channel due to strong up to $\sim 8 \times 10^{-3}$ ($\sim 0.27$ μm/s) horizontal flow directed in the negative sense (see Figure 25a), whereas the vertical flow $w$ (see Figure 25b) is characterized by very small value $\sim 4 \times 10^{-4}$ ($\sim 13.2$ nm/s) directed in the opposite sense ref. [1].

B. Fast heating mode when a laser beam is focused inside of an HAN channel

Now we consider the case when the hybrid aligned nematic microvolume is heated by the laser beam focused in the interior ($x = 0.0$ and $z = 0.97$) of the HAN channel, at different times $\tau_1 = 10^{-8}$ ($\sim 1.5$ ns) [curve (1)], $\tau_2 = 4 \times 10^{-8}$ ($\sim 6.0$ ns) [curve (2)], $\tau_3 = 7 \times 10^{-8}$ ($\sim 10.5$ ns) [curve (3)], and $\tau_{in} = \tau_4 = 10^{-7}$ ($\sim 15.0$ ns) [curve (4)], respectively. In this case, the value of the dimensional heat flux coefficient is equal to $\mathcal{O}_0 \sim 14.0$ $W$, whereas the heating occurs during time $t_{in} \sim 15.0$ ns. Figure 27 shows how three vortices can be maintained in the HAN channel with a free LC/vacuum interface $\Gamma$, during the fast heating mode, one biggest vortical flow in the vicinity of the heat source and directed in the negative sense (anticlockwise) around their center $x = 0.0$, $z \sim 0.93$, and two smallest vortices, which are settled down close to the points $x = \pm 0.13$ and $z \sim 0.93$ [11], respectively. These calculations also show that the range of distance $z$, counted from the lower boundary of the HAN channel, over which the laser beam cannot disturb the nematic phase, is $0.8 \leq z \leq 1.0$, i.e., which is approximately 80% of the LC sample (see Figure 27). Notice that the duration of the energy injection $\tau_{in}$ into the LC sample is restricted only by the nematic phase stability. Further calculations (cooling mode), based on the nonlinear extension of the Ericksen–Leslie theory, show that the nematic material settles down to the rest during the time term $\tau_8 \sim 2.56$ ($\sim 0.5$ s), after switching off the laser power, where both the horizontal $u$ and vertical $w$ components of the velocity $\mathbf{v}$ are equal to zero, and the temperature field $\chi$ across the HAN channel finally downfalls to the value on the lower and two lateral boundaries.

The results presented in [11–13] indicate that with a change in the nature of laser radiation focusing, from the surface of the nematic channel to its volume, leads to a significant change in hydrodynamic flows. The question of how the depth of laser radiation focus affects the nature of hydrodynamic flows in the hybrid aligned nematic channel will be discussed in the next paragraph.

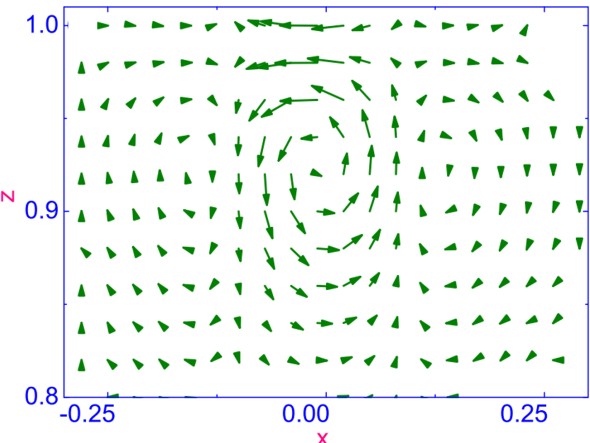

**Figure 27.** (Fast heating mode). Distribution of the velocity field $\mathbf{v} = u\hat{\mathbf{i}} + w\hat{\mathbf{k}} = -\nabla \times \hat{\mathbf{j}}\psi$ in the HAN channel after heating during $\tau = \tau_{in}$ [12]. Here, 1 mm of the arrow length is equal to 0.4 μm/s.

### 2.5. How the Depth of Laser Radiation Focus Affects the Nature of Hydrodynamic Flows in Nematic Channel

In this paragraph, we will consider the effect of the depth of focus of laser radiation in the microscopic volume of the HAN channel with the free LC/air interface on the nature of formation of the hydrodynamic flow and temperature distribution [11,12].

Figures 28 and 29 show the distribution of the dimensionless temperature $\chi(\tau, x = 0.0, z)$ along the $z$-axis ($-1.0 \le z \le 1.0$), when the laser beam is focused in the center ($x = 0.0$) of the HAN channel, at different depths [11,12]: $z_0 = 0.80$ (see Figure 28a), $z_0 = 0.90$ (see Figure 28b), $z_0 = 0.94$ (see Figure 29a), and $z_0 = 0.98$ (see Figure 29b), respectively.

The heating mode when the laser beam is focused in the interior of the HAN channel is given at different times $\tau_i = 2^i \times 10^{-5}$ ($i = 6, \ldots, 10$), respectively. It has been shown that as the focus of the laser beam is shifted in the depth of the nematic microvolume, the temperature profiles across the nematic channel do not undergo the crucial change. For instance, in the case when the laser beam is focused on the maximum depth ($z_0 = 0.8$), the heating does not reach the LC/air interface (see Figure 28a [11,12]).

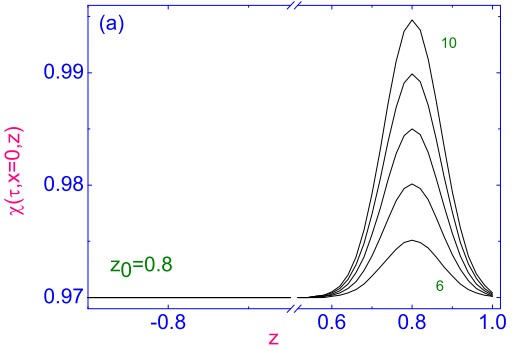
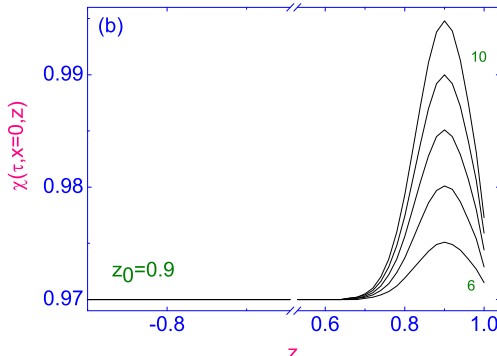

**Figure 28.** Distribution of the dimensionless temperature $\chi(\tau, x = 0.0, z)$ along the $z$-axis ($-1.0 \le z \le 1.0$), when the laser beam is focused in the center ($x = 0.0$) of the HAN channel, at different depths [11,12]: (**a**) $z_0 = 0.80$, and (**b**) $z_0 = 0.90$, respectively. The numbering of the curves increases from $i = 6$ to $i = 10$.

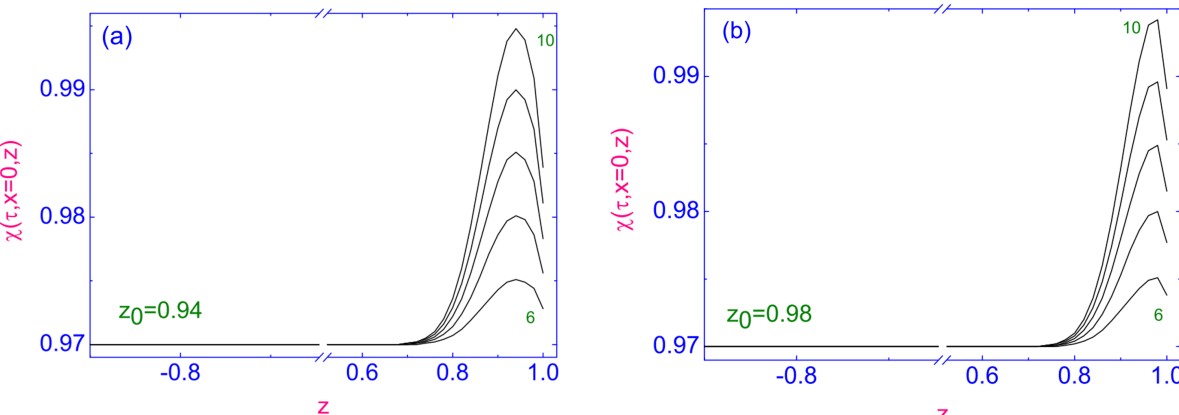

**Figure 29.** The same as in Figure 28a, but the distribution of the dimensionless temperature $\chi(\tau, x = 0, z)$ along the $z$-axis $(-1.0 \leq z \leq 1.0)$ is given at different depths [11,12]: (**a**) $z_0 = 0.94$, and (**b**) $z_0 = 0.98$, respectively.

In turn, the velocity profiles across the HAN channel undergo the crucial change. Figures 30–33 show the distribution of the horizontal $u(\tau, x = 0.0, z)$ and vertical $w(\tau, x = 0.0, z)$ components of the velocity vector $\mathbf{v} = u\hat{\mathbf{i}} + w\hat{\mathbf{k}} = -\nabla \times \hat{\mathbf{j}}\psi$ along the $z$-axis $(-1.0 \leq z \leq 1.0)$, when the laser beam is focused in the center $(x = 0.0)$ of the HAN channel, at different depths. Figure 30a,b [11,12] show the distribution of the horizontal $u(\tau, x = 0.0, z)$ component of the velocity $\mathbf{v}$ along the $z$-axis $(-1.0 \leq z \leq 1.0)$, when the laser beam is focused in the center $(x = 0.0)$ of the HAN channel, but at different depths [11,12]: Figure 30a, at $z_0 = 0.80$ and Figure 30b, at $z_0 = 0.90$, respectively. In turn, the Figure 31a,b show the distribution of the horizontal $u(\tau, x = 0.0, z)$ component of the velocity $\mathbf{v}$ along the $z$-axis $(-1.0 \leq z \leq 1.0)$, at different depths [11,12], Figure 31a, at $z_0 = 0.94$, Figure 31b, at $z_0 = 0.98$, respectively. Figure 32a,b show the distribution of the vertical $w(\tau, x = 0.0, z)$ component of the velocity field $\mathbf{v}$ along the $z$-axis $(-1.0 \leq z \leq 1.0)$, when the laser beam is focused in the center $(x = 0.0)$ of the HAN channel, at different depths: Figure 32a, at $z_0 = 0.80$, Figure 32b, at $z_0 = 0.90$, respectively. In turn, Figure 33a,b show the distribution of the vertical $w(\tau, x = 0.0, z)$ component of the velocity $\mathbf{v}$ along the $z$-axis $(-1.0 \leq z \leq 1.0)$, at different depths, Figure 33a, at $z_0 = 0.94$, and Figure 33b, at $z_0 = 0.98$, respectively.

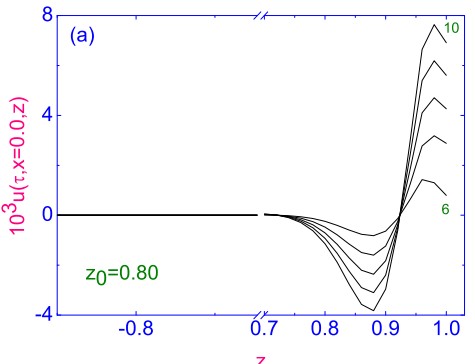
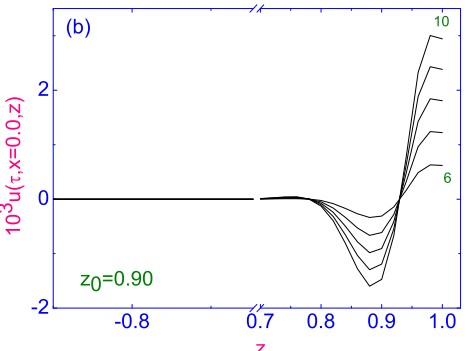

**Figure 30.** Distribution of the horizontal $u(\tau, x = 0.0, z)$ component of the velocity $\mathbf{v}$ along the $z$-axis $(-1.0 \leq z \leq 1.0)$, when the laser beam is focused in the center $(x = 0.0)$ of the HAN channel, at different depths [11,12]: (**a**) $z_0 = 0.80$ and (**b**) $z_0 = 0.90$, respectively. The numbering of the curves increases from $i = 6$ to $i = 10$.

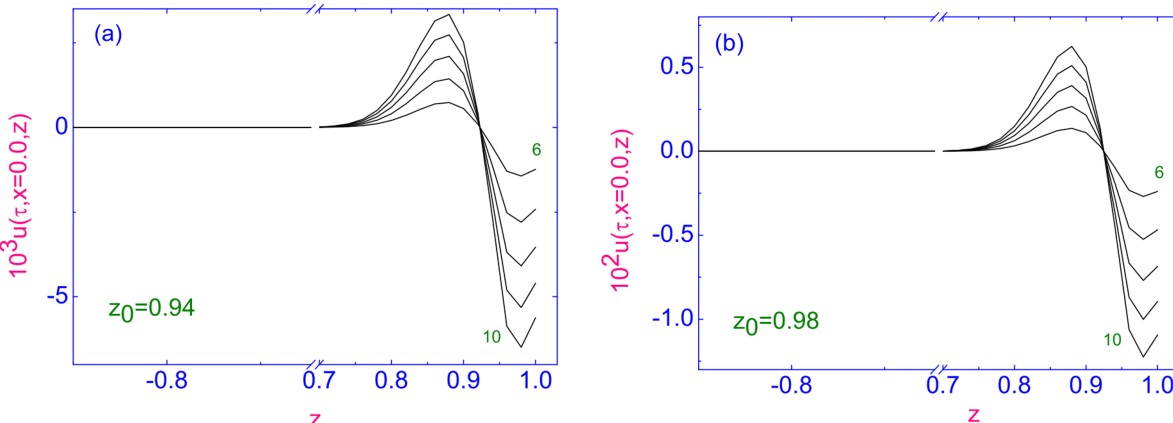

**Figure 31.** The same as in Figure 30a, but the distribution of the horizontal $u(\tau, x = 0.0, z)$ component of the velocity **v** along the $z$-axis ($-1.0 \leq z \leq 1.0$) is given at different depths [11,12]: (**a**) $z_0 = 0.94$, and (**b**) $z_0 = 0.98$, respectively.

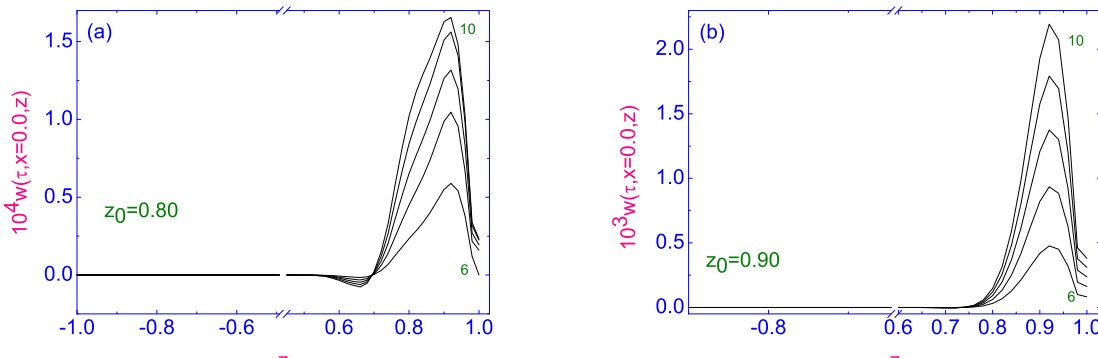

**Figure 32.** Distribution of the vertical $w(\tau, x = 0.0, z)$ component of the velocity **v** along the $z$-axis ($-1.0 \leq z \leq 1.0$), when the laser beam is focused in the center ($x = 0.0$) of the HAN channel, at different depths [11,12]: (**a**) $z_0 = 0.80$ and (**b**) $z_0 = 0.90$, respectively. The numbering of the curves increases from $i = 6$ to $i = 10$.

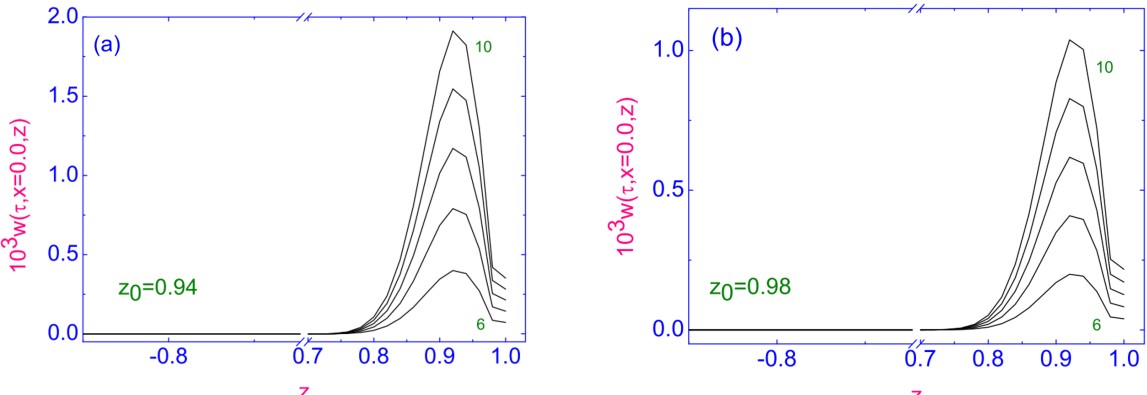

**Figure 33.** The same as in Figure 32a, but the distribution of the vertical $w(\tau, x = 0.0, z)$ component of the velocity **v** along the $z$-axis ($-1.0 \leq z \leq 1.0$) is given at different depths [11,12]: (**a**) $z_0 = 0.94$, and (**b**) $z_0 = 0.98$, respectively.

It has been shown that as the focus of the laser beam is shifted in the depth of the HAN channel in the vicinity of the LC/air interface, the horizontal component of the velocity $u(\tau, x = 0, z)$ changes its direction from negative to positive, approximately at the point $x_0 = 0.0, z_0 \sim 0.9$, whereas the vertical component of the velocity $w(\tau, x = 0, z)$ rapidly drops to zero. It should be noted that the greatest value of $u(\tau, x = 0, z)$, directed in the positive sense in the vicinity of the LC/air interface, is achieved in the case when the

laser beam is focused on the maximum depth of penetration ($z_0 = 0.8$) in the LC volume, whereas the greatest value of $u(\tau, x = 0, z)$, directed in the negative sense in the vicinity of the LC/air interface, is achieved in the case when the laser beam is focused on the minimum depth of penetration ($z_0 = 0.98$) in the LC volume. In both of these cases, the vertical component of the velocity vector $w(x = 0, z, \tau)$ at the free LC/air interface is almost zero. Therefore, this distribution of components of the velocity field shows that in the area close to the LC/air interface ($0.8 < z < 1.0$), the vortex flow is excited due to the energy pumping by laser radiation, similar to what is shown in Figure 26. These calculations [11,12] also show that with further penetration of the injecting energy to the bulk of the LC phase, from $x_0 = 0.0, z_0 = 0.98$ to $x_0 = 0.0, z_0 = 0.8$, the thermally excited vortical flow changes the direction from anticlockwise, around the point $x = 0.0, z = 0.98$, to clockwise, around the point $x = 0.0, z = 0.8$, approximately at the point $x_0 = 0.0, z_0 \sim 0.9$.

Based on the numerical results describing the formation of hydrodynamic flows in microsized HAN channels with free surfaces under the effect of laser radiation focused in the volume of the LC phase, the following conclusion can be made. First, the calculations, based on the appropriate nonlinear extension of the classical Ericksen–Leslie theory, show that due to the interaction between $\nabla T$ and the gradient of the director field $\nabla \hat{\mathbf{n}}$, in the nematic volume the thermally excited three-vortical fluid flow is maintained. Second, the direction and magnitude of the hydrodynamic flow at a fixed time of pumping energy and the laser output power are affected by the depth of the laser injection. The above calculations also show that the range of distances measured from the lower solid boundary at which the laser beam cannot disturb the nematic phase is approximately 80% of the nematic sample.

It should be noted that the vortical flow in homeotropically oriented LC film doped by chiral molecules, using the circular polarization techniques, recently has been observed [10]. It has been shown that at the beginning of laser irradiation, the thermocapillary radial flow is transformed into a circular flow around the position of the laser spot on the free LC interface. The formation of the circular flow on the top of the LC film has been ascribed to thermocapillary convection in the LC sample.

In turn, in the above-mentioned cases [11,12], the vortical flow occurred in the vicinity of the free LC/air interface and penetrated to the bulk of the LC sample. The mechanism responsible for the occurrence of the vortical flow near the LC/air interface is based on the coupling between the director and the temperature gradients initiated by the laser beam radiation. Thus, this vortical flow is a unique phenomenon that is exhibited only by liquid crystal systems, and is expected to be applied to new optical-thermal tweezers [1].

## 3. Heat Driven Nematic Flow in Cylindrical Microfluidic Channel

The objective of this section is to analyze the response of the nematic phase confined in the cylindrical micro cavity between two horizontal coaxial cylinders under the influence of the temperature gradient $\nabla T$ directed from the inner (outer) cooler (warmer) to outer (inner) warmer (cooler) cylinders [28]. Therefore, we are primarily interested here in describing how the temperature gradient across the microvolume cavity between two coaxial cylinders can produce the hydrodynamic flow $\mathbf{v}$. It has been treated in the framework of the classical Ericksen–Leslie theory [16,17], supplemented by the thermomechanical correction of shear stress [7,15], and the entropy balance equation [18]. The hydrodynamic model in which the above problem will be considered is the same as in Section 2.1, and is based on the interaction effect of the director $\hat{\mathbf{n}}$ and temperature $T$ gradients with the velocity $\mathbf{v}$ field. The magnitude of the hydrodynamic flow is proportional to $v \sim \frac{d}{\eta} \sigma_{zr}^{tm}$, where $\sigma_{zr}^{tm}$ is the tangential component of the thermomechanical stress tensor, $\eta$ is the viscosity of the nematic phase, and $d$ is the thickness of the hybrid aligned nematic (HAN) cavity confined between two infinitely long horizontal coaxial cylinders. It will be shown below that the tangential component of the thermomechanical stress tensor plays a crucial role in the formation of the thermally excited vortical flow in the HAN cavity.

To fix ideas and notations, in ref. [28] was considered the HAN system composed of asymmetric polar molecules, such as *cyanobiphenyls*, at the density $\rho$, and confined between two infinitely long horizontal coaxial cylinders with radii $R_1$ and $R_2$. Here, $R_1 < R_2$, that imposes a preferred orientation of the average molecular direction $\hat{\mathbf{n}}$ on both bounding surfaces, for instance, homeotropic on the inner cooler ($T_{in} = T_1$), and planar, on the outer warmer ($T_{out} = T_2$) bounding cylinders. Therefore, it was considered the HAN system under the influence of the radially directed temperature gradient $\nabla T$ parallel to the unit cylindrical vector $\hat{\mathbf{e}}_r$ along the radius $r$ [28]. The coordinate system defined by this geometry assumes that the director $\hat{\mathbf{n}}$ lies in the $rz$ plane, where $\hat{\mathbf{e}}_z$ is the unit vector which coincides with the planar director orientation, for instance, on the outer warmer (inner cooler) bounding cylinder ($\hat{\mathbf{e}}_z \parallel \hat{\mathbf{n}}_{r=R_2(=R_1)}$), $\hat{\mathbf{e}}_r$ denotes the unit vector along the radius $r$, and $\hat{\mathbf{e}}_\varphi = \hat{\mathbf{e}}_z \times \hat{\mathbf{e}}_r$ is the tangential unit vector.

Assuming that the temperature gradient $\nabla T$ varies only in the $r$ direction, $\nabla T = \frac{\partial T(t,r)}{\partial r}\hat{\mathbf{e}}_r$, and the director $\hat{\mathbf{n}}$ belongs to the $rz$ plane. Further, it was proposed that in the case of the infinitely long cylinders, the components of the director $\hat{\mathbf{n}} = \sin\theta(t,r)\hat{\mathbf{e}}_r + \cos\theta(t,r)\hat{\mathbf{e}}_z$, as well as the rest of the relevant physical quantities depend only on the radius $r$ and on the time $t$. Here $\theta$ denotes the angle between the direction of the director $\hat{\mathbf{n}}$ and the unit vector $\hat{\mathbf{e}}_z$ directed parallel to the long cylinder's axis. Moreover, it was assumed the homeotropic and homogeneous strong anchoring conditions on the outer and inner cylinders [28]

$$\theta(r)_{r=R_2} = 0, \theta(r)_{r=R_1} = \frac{\pi}{2}, \tag{36}$$

and no-slip boundary conditions for the nematogenic molecules on both bounding cylindrical surfaces, i.e.,

$$v(r)_{r=R_1} = v(r)_{r=R_2} = 0, \tag{37}$$

respectively. Upon assuming an incompressible fluid $\nabla \cdot \mathbf{v} = 0$, the hydrodynamic equations describing the reorientation of the HAN system confined in the microvolume between two horizontal coaxial cylinders under the influence of the temperature gradient $\nabla T$ directed from the inner cooler to outer warmer cylinders

$$T(r)_{r=R_1} = T_{in}, T(r)_{r=R_2} = T_{out}, \tag{38}$$

can be derived from the balance of elastic, viscous, and thermomechanical torques $\mathbf{T}_{el} + \mathbf{T}_{vis} + \mathbf{T}_{tm} = 0$, the Navier–Stokes equation for the velocity field $\mathbf{v}$, excited by $\nabla T$, and the equation for heat conduction. Here, $\mathbf{T}_{el} = \frac{\delta F_{el}}{\delta \hat{\mathbf{n}}} \times \hat{\mathbf{n}}$, $\mathbf{T}_{vis} = \frac{\delta \mathcal{R}_{vis}}{\delta \dot{\hat{\mathbf{n}}}} \times \hat{\mathbf{n}}$, $\mathbf{T}_{tm} = \frac{\delta \mathcal{R}_{tm}}{\delta \dot{\hat{\mathbf{n}}}} \times \hat{\mathbf{n}}$, $\dot{\hat{\mathbf{n}}} = \frac{d\hat{\mathbf{n}}}{dt}$ is the material derivative of the director $\hat{\mathbf{n}}$, $F_{el} = \frac{1}{2}\left(K_1(\nabla \cdot \hat{\mathbf{n}})^2 + K_3(\hat{\mathbf{n}} \times \nabla \times \hat{\mathbf{n}})^2\right)$ is the elastic energy, and $\mathcal{R} = \mathcal{R}_{vis} + \mathcal{R}_{tm} + \mathcal{R}_{th}$ is the full dimensionless Rayleigh dissipation function composed by the viscous, thermomechanical, and thermal contributions, respectively. The incompressibility condition

$$\nabla \cdot \mathbf{v} = \frac{1}{r}\frac{\partial(rv_r(t,r))}{\partial r} + \frac{\partial v_z(t,r)}{\partial z} = \frac{1}{r}\frac{\partial(rv_r(t,r))}{\partial r} = 0,$$

together with the no-slip condition (37) implies the existence only one nonzero component for the vector $\mathbf{v}$, viz. $\mathbf{v}(t,r) = v_z(t,r)\hat{\mathbf{e}}_z \equiv u(t,r)\hat{\mathbf{e}}_z$. In the cylindrical coordinate system the dimensionless torque balance equation takes the form [28]

$$\overline{\gamma}_1(\chi)\theta_{,\tau} = \mathcal{A}(\theta)u_{,r} + (\mathcal{G}(\theta)\theta_{,r})_{,r} - \frac{1}{2}\mathcal{G}_{,\theta}(\theta)\theta_{,r}^2 + \frac{\theta_{,r}}{r}\mathcal{G}(\theta)$$

$$+ \frac{1}{2}\left(\frac{\overline{K}_1(\chi)}{r}\right)_{,r} - \delta_6\chi_{,r}\left[\theta_{,r}\left(\frac{1}{2} + \cos^2\theta\right) + \frac{3}{4r}\sin 2\theta\right], \tag{39}$$

where $\overline{\gamma}_1 = \gamma_1(\chi)/\gamma_{10}$, $\delta_6 = \xi\frac{T_{NI}}{K_{10}}$ is the parameter of the nematic system, $\theta_{,\tau} = \frac{\partial\theta(\tau,r)}{\partial\tau}$, $\theta_{,r} = \frac{\partial\theta(\tau,r)}{\partial r}$, $u_{,r} = \frac{\partial u(\tau,r)}{\partial r}$, and $\mathcal{A}(\theta) = -\frac{1}{2\gamma_{10}}[\gamma_1(\chi) + \gamma_2(\chi)u_{,r}(\tau,r)\cos 2\theta]$ and $\mathcal{G}(\theta) = \frac{K_1(\chi)}{K_{10}}\sin^2\theta + \frac{K_3(\chi)}{K_{10}}\cos^2\theta$ are the hydrodynamic and elastic functions, and $\chi(\tau,r) = T(\tau,r)/T_{NI}$ is the dimensionless temperature. Here, $\gamma_{10}$ and $K_{10}$ are the highest values of the RVC $\gamma_1(\chi)$ and of the splay constant $K_1(\chi)$ in the temperature interval $\Delta\chi = \chi_2 - \chi_1$ belonging to the nematic phase, $\chi_1 = T_1/T_{NI}$, $\chi_2 = T_2/T_{NI}$, $\tau = \left(\frac{K_{10}}{\gamma_{10}d^2}\right)t$ is the dimensionless time, $\overline{r} = \frac{r}{d}$ is the dimensionless radius, and $d = R_2 - R_1$ is the capillary gate, respectively.

Notice that the overbars in the space variable $r$ have been (and will be) eliminated in the last as well as in the following equations. In the case of incompressible fluid, the dimensionless Navier–Stokes equation (in cylindrical coordinates) takes the form [28]

$$\delta_7 u_{,\tau} = \nabla_{,r} \cdot \sigma_{rz}, \tag{40}$$

where $\delta_7 = \frac{\rho K_{10}}{\gamma_{10}^2}$ is an additional parameter of the system, $\nabla_{,r}\cdot\sigma_{rz} = \frac{\partial\sigma_{rz}}{\partial r} + \frac{\sigma_{rz}}{r}$, and $\sigma_{rz} = \frac{\partial\mathcal{R}}{\partial u_{,z}}$ is the tangential ST $\sigma_{ij}$ $(i,j = r,z)$ component. Here, $\mathcal{R}(\tau,r) = \frac{\gamma_{10}d^4}{K_{10}}\mathcal{R}(t,r)$ is the full dimensionless Rayleigh dissipation function, where $\mathcal{R}(t,r) = \mathcal{R}_{vis} + \mathcal{R}_{tm} + \mathcal{R}_{th}$, and $\mathcal{R}_{vis} = \frac{1}{2}\left[\gamma_1\theta_{,t}^2 + u_{,r}\theta_{,t}\overline{\mathcal{A}}(\theta) + h(\theta)u_{,r}^2\right]$ is the viscous, $\mathcal{R}_{tm} = \xi T_{,r}\left[\theta_{,t}\theta_{,r}(\frac{1}{2} + \cos^2\theta) + \frac{1}{4r}\theta_{,t}\sin 2\theta + u_{,r}\mathcal{H}(\theta)\right]$ is the thermomechanical, and $\mathcal{R}_{th} = \frac{T_{,r}^2}{2T}\left(\lambda_\parallel\sin^2\theta + \lambda_\perp\cos^2\theta\right)$ is the thermal contribution, respectively. Here, $\overline{\mathcal{A}}(\theta) = \gamma_{10}\mathcal{A}(\theta)$, $4h(\theta) = 2\gamma_1(T) + \alpha_5(T) + \alpha_6(T) + 2\alpha_4(T) + 2\gamma_2(T)\cos 2\theta + \alpha_1(T)\sin^2 2\theta$ and $\mathcal{H}(\theta) = \theta_{,r}\cos^2\theta(1 + \frac{1}{2}\sin^2\theta) + \frac{3}{4r}\sin 2\theta$ are both hydrodynamic functions, and $\alpha_i(T)$ $(i = 1,\ldots,6)$ are the six temperature dependent Leslie coefficients. When a small temperature gradient $\nabla T$ (in our case $\sim 0.1$ K/$\mu$m) is set up across the cavity between two infinitely long coaxial cylinders, one expects the temperature field $\chi(\tau,r)$ to satisfy the dimensionless heat conduction equation [28]

$$\delta_8\chi_{,\tau} = \frac{1}{r}\left[r\chi_{,r}\left(\lambda\sin^2\theta + \cos^2\theta\right)\right]_{,r} +$$
$$\delta_9\frac{1}{r}\left[r\chi\left(\theta_{,\tau}\theta_{,r}(\frac{1}{2} + \cos^2\theta) + \frac{\theta_{,\tau}}{4r}\sin 2\theta + u_{,r}\frac{\mathcal{H}(\theta)}{d}\right)\right]_{,r}, \tag{41}$$

where $\lambda = \frac{\lambda_\parallel}{\lambda_\perp}$, $\delta_8 = \frac{\rho C_p K_{10}}{\lambda_\perp\gamma_{10}}$, and $\delta_9 = \frac{\xi K_{10}}{\lambda_\perp\gamma_{10}d^2}$ are the two additional parameters of the system. The above-mentioned hybrid anchoring conditions for the director $\hat{n}$ now read

$$\theta_{r=a} = \frac{\pi}{2}, \theta_{r=a+1} = 0, \tag{42}$$

$$\theta_{r=a} = 0, \theta_{r=a+1} = -\frac{\pi}{2}, \tag{43}$$

whereas the velocity on these cylinders has to satisfy the no-slip boundary condition

$$u(r)_{r=a} = u(r)_{r=a+1} = 0. \tag{44}$$

As for the temperature field $\chi(\tau,r)$, the corresponding boundary conditions read

$$\chi_{r=a} = \chi_1, \chi_{r=a+1} = \chi_2, \tag{45}$$

$$\chi_{r=a} = \chi_2, \chi_{r=a+1} = \chi_1, \tag{46}$$

respectively. Here, $a = \frac{R_1}{R_2 - R_1}$ is the dimensionless size of the HAN capillary.

Notice that this approach is only valid for the nematic phase. For the case of $4 - n - pentyl - 4' - cyanobiphenyl$ (5CB), at temperature corresponding to the nematic phase,

the parameters involved in Equations (39)–(41) are $\delta_6 \sim 32$, $\delta_7 \sim 10^{-6}$, $\delta_8 \sim \times 10^{-3}$, and $\delta_9 \sim 10^{-13}$ (for details, see [28]). Using the fact that $\delta_7, \delta_8$, and $\delta_9 \ll 1$, both the Navier–Stokes (40) and the heat conduction (41) equations can be considerably simplified. Thus, the whole left-hand side of Equations (40) and (41) can be neglected, reducing it to

$$\sigma_{rz} = \frac{\mathcal{C}(\tau)}{r}, \tag{47}$$

and

$$\left[ r\chi_r \left( \lambda \sin^2 \theta + \cos^2 \theta \right) \right]_r = 0, \tag{48}$$

where the functions $\mathcal{C}(\tau)$ do not depends on $r$ and will be fixed by the boundary condition Equation (44). The last equation has a solution

$$\chi(\tau, r) = \frac{\Delta\chi}{\mathcal{I}} \int_a^r \frac{dr}{r(\lambda \sin^2 \theta + \cos^2 \theta)} + \chi_i, \tag{49}$$

where $\mathcal{I} = \int_a^{a+1} \frac{dr}{r(\lambda \sin^2 \theta + \cos^2 \theta)}$, and $\chi_i$ ($i = 1, 2$).

The evolution of the director $\hat{\mathbf{n}}$ to its equilibrium orientation $\hat{\mathbf{n}}_{eq}$, which is described by the angle $\theta(\tau, r)$, from the initial condition $\theta(\tau = 0, a < r \leq a + 1) = \theta_{el}(r)$ to $\theta_{eq}(r)$ (see Figure 34a), and velocity $u(\tau, r) = v_z(\tau, r)$ (see Figure 34b), in the HAN capillary, at different times: $\tau_1 = 0.02$ ($\sim 1.4$ s) (curve (1)) to $\tau_{10} = \tau_R = 0.2$ ($\sim 14$ s) (curve (10)), are shown in Figure 34 [28].

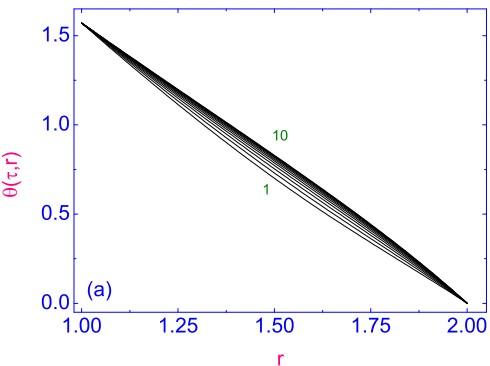 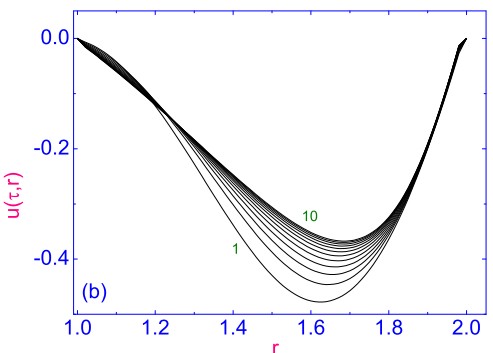

**Figure 34.** (**a**) The distance $r$ dependence of the angle $\theta(\tau, r)$ across the HAN cavity between two $a \leq r \leq a + 1$ infinitely long coaxial cylinders, under the influence of the temperature gradient $\nabla\chi$, directed from the cooler inner ($T_{in} = 0.97$) to warmer outer ($T_{out} = 0.9862$) cylinders, at different times $\tau_k = \frac{k}{10}\tau_R$ ($k = 1, \ldots, 10$), whose values increase from curve (1) to curve (10) [28]. Here $\tau_R = 0.2$. (**b**) The same as in (**a**), but the distance $r$ dependence of the velocity $u(\tau, r)$. All calculations were carried out for $a = 1.0$.

In the reviewed case [28], the HAN capillary is heated from above with the dimensionless temperature difference $\Delta\chi = 0.0162$ ($\sim 5$ K). The solution of the system of nonlinear partial differential equations Equations (39)–(41), together with the boundary conditions (42)–(46), has been obtained by means of the numerical relaxation method [24]. The relaxation criterion $\epsilon = |(\theta_{(m+1)}(\tau, r) - \theta_{(m)}(\tau, r))/\theta_{(m)}(\tau, r)|$ was chosen to be $10^{-4}$, and the numerical procedure was then carried out until a prescribed accuracy was achieved. Here, $m$ is the iteration number, and $\tau_R$ is the relaxation time of the HAN system.

In turn, the evolution of the temperature field $\chi(\tau, r)$ to its equilibrium distribution $\chi_{eq}(r) = \chi(\tau_R, r) = \chi(\tau_{10}, r)$ across the cylindrical cavity $a < r \leq a + 1$, at different times, from $\tau_1 = 0.02$ ($\sim 1.4$ s) (curve (1), to $\tau_{10} = \tau_R = 0.2$ ($\sim 14$ s) (curve (10)), is shown in Figure 35 [28].

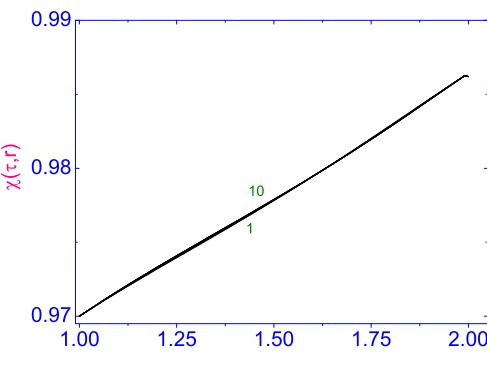

**Figure 35.** The same as in Figure 34a, but the distance $r$ dependence of the temperature $\chi(\tau, r)$ [28]. All calculations were carried out for $a = 1.0$.

In that case, the lower cooler surface is kept at constant temperature $\chi_1 = 0.97$ ($T_1 \sim 298$ K), and the relaxation of the temperature field $\chi(\tau, r)$ across the HAN cavity is characterized practically by the linear increasing of the values of $\chi(\tau, r)$, from $\chi_1$ to $\chi_2$ (see Figure 35).

According to these calculations [28], the evolution of the dimensionless velocity $u(\tau, r)$ in the HAN cavity between two infinitely long coaxial cylinders is characterized by the monotonic decrease of $|u(\tau, r)|$ upon increasing $\tau$, before getting to the equilibrium distributions $u_{eq}(r) = u(\tau_R, r) = u(\tau_{10}, r)$ across the HAN cavity. That distribution is characterized by the minimum near the middle part of the cavity, where the hydrodynamic flow is directed in the negative sense (see Figure 34b, curve (10)).

It should be noted that the equilibrium hydrodynamic flow $u_{eq}(r)$ change direction, from the negative to positive sense, across the full thickness of the HAN cavity, after changing the temperature gradient $\nabla \chi$ direction, from the cooler ($T_{in} = 298$ K) inward to the warmer outward ($T_{out} = 303$ K) cylinders on the the warmer ($T_{in} = 303$ K) inward to the cooler outward ($T_{out} = 298$ K) (see Figure 36 and Figure 37, curves (2) and (1), respectively.).

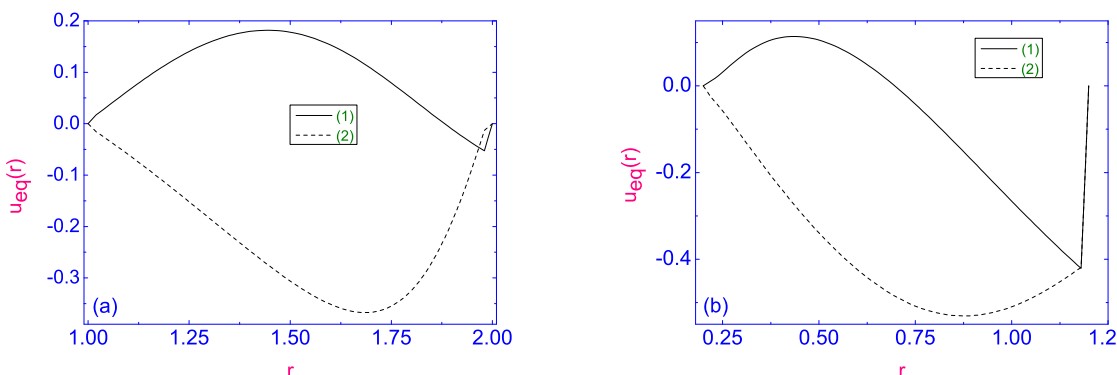

**Figure 36.** The distance $r$ dependence of the equilibrium velocity $u_{eq}(r)$ across the nematic cavity between two $a \leq r \leq a + 1$ infinitely long coaxial cylinders, with the anchoring hybrid condition in the form of Equation (42), under influence of the $\nabla \chi$, directed from the cooler (warmer) inward $\chi_1$ ($\chi_2$) to warmer (cooler) outward $\chi_2$ ($\chi_1$) cylinders (see curves (2) and (1), respectively), calculated for a number of values of $a$ [28]: (**a**) 1.0 and (**b**) 0.2, respectively.

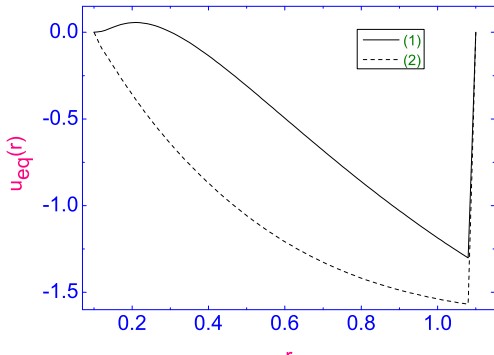

**Figure 37.** The same as in Figure 36, but the value of *a* is equal to 0.1 [28].

These calculations [28] show that increasing of the nematic cavity size $d \equiv R_2 - R_1$ between two infinitely long coaxial cylinders, when the temperature gradient $\nabla \chi$ is directed from the inward to outward cylinders, leads to increase of the maximum of the absolute magnitude of the dimensionless velocity $u_{eq}(r)$, from $u_{eq}^{max} \sim 0.4$ (~11 μm/s), at $a = 1$ ($R_2 = 2R_1$) (see Figure 36a) to $u_{eq}^{max} \sim 1.5$ (~41.3 μm/s), at $a = 0.1$ ($R_2 = 11R_1$) (see Figure 37), respectively.

According to these calculations [28], in the case when the hybrid oriented anchoring conditions for director is described both by Equations (42) and (43), with $(\chi_1)_{r=a} > (\chi_2)_{r=a+1}$, one has arrived to the picture where the nematic fluid settles down to the stationary flow regime in the negative verse (see Figures 36–39).

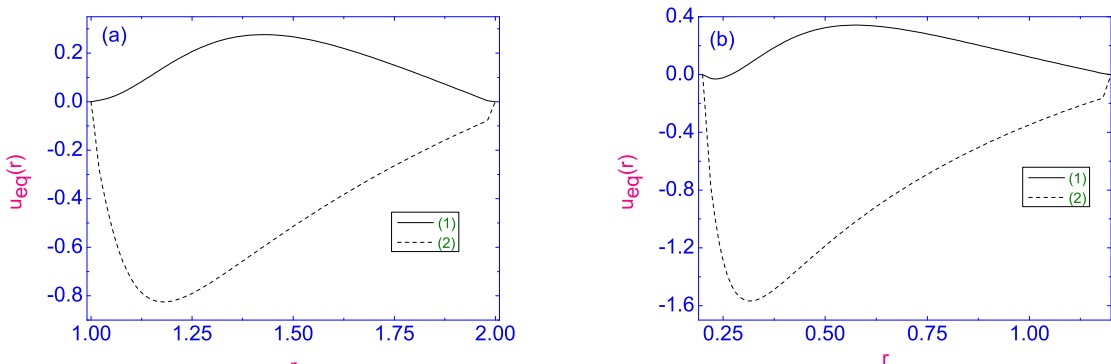

**Figure 38.** The same as in Figure 36, but the anchoring hybrid condition is set up in the form of Equation (43) [28]. (**a**) 1.0 and (**b**) 0.2, respectively.

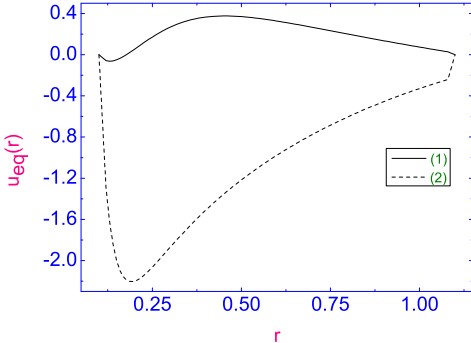

**Figure 39.** The same as in Figure 38, but for a value of *a* equal to 0.1 [28].

In the first case (Equation (42)), the maximum of the velocity field $u_{eq}^{max}(r)$ is found in the vicinity of the cooler outward cylinder (see Figure 37), whereas in the second case (Equation (43)), the maximum of $u_{eq}^{max}(r)$ is found in the vicinity of the warmer inward cylinder (see Figure 39), respectively. Physically, this means that the thermomechanical force can overcome the viscous and elastic forces, and the nematic drop, confined between two infinitely long hybrid-oriented cylinders, begins to move in the horizontal direction with the stationary distributed velocity across the nematic cavity. Thus, we arrive at the picture where there is a balance between the applied temperature gradient and the viscous, elastic, and anchoring forces, and, in general, the nematic fluid settles in a stationary flow regime along the long axis of the cylinder's. The magnitude of the hydrodynamic flow $u_{eq}$ is proportional to the tangential component of the thermomechanical stress tensor $\sigma_{zr}^{tm}$ and the cavity size $R_2 - R_1$, and the direction of $u_{eq}$ is influenced by both the direction of the heat flux and the character of the preferred anchoring of the director to the restricted cylinders.

The highest value of the dimensionless velocity $u_{eq}^{max}(r)$ is built up in the hybrid aligned nematic cavity, when the temperature gradient $\nabla \chi$ directed from the outward cooler ($T_{out} = 298$ K) to inward warmer ($T_{in} = 303$ K) cylinders, in the vicinity of the homogeneously aligned inward cylinder (see Equation (43)), and equal to $\sim$2 ($\sim$55 µm/s) (see Figure 39), and is directed in the negative sense.

It has been found in ref. [28] that the size of the HAN cavity $a = \frac{R_1}{R_2 - R_1}$ has a pronounced effect on the magnitude of $u_{eq}^{max}(a)$ (see Figure 40).

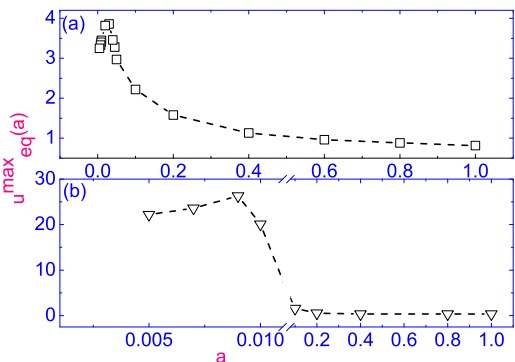

**Figure 40.** Dependence of $u_{eq}^{max}(a)$ on the dimensionless size of the nematic cavity $a = \frac{R_1}{R_2 - R_1}$, with the anchoring hybrid condition in the form of Equation (43), for two cases: first, (**a**) when the heating mode is directed from inward to outward bounding cylinders ($\chi_{r=a+1} > \chi_{r=a}$), second, (**b**) when the heating mode is directed from outward to inward bounding cylinders ($\chi_{r=a} > \chi_{r=a+1}$) [28], respectively.

Figure 40a shows the effect of $a$ on the magnitude of the highest dimensionless velocity $u_{eq}^{max}(a)$, when the temperature on the inward cylinder $\chi_{r=a}$ is greater than the temperature on the outward cylinder $\chi_{r=a+1}$. In the case of $a = 0.03$, i.e., when the radius of the outward cylinder $R_2 \sim 34R_1$, the maximum value of the dimensionless equilibrium velocity has the biggest value which is equal to $\sim$3.82 ($\sim$105 µm/s), and is directed in the negative sense. Note that the further decrease of $a$ leads to decreasing of the highest dimensionless velocity $u_{eq}^{max}(a)$. Figure 40b shows the effect of $a$ on the magnitude of the highest dimensionless velocity $u_{eq}^{max}(a)$, in the case when $\chi_{r=a+1} > \chi_{r=a}$. Here, the biggest value is equal to $\sim$26 ($\sim$0.7 mm/s), at $a = 0.01$ ($R_2 \sim 112R_1$), and $\Delta\chi = 0.0162$ ($\sim$5 K), and directed also in the negative sense.

It should be noted here that the highest temperature difference $\Delta = T_{in(out)} - T_{out(in)}$ across the nematic cavity is still such that both $T_{in}$ and $T_{out}$ fall within the stability region of the nematic phase. The temperature difference, for instance, in $\sim$5 K in the nematic cavity confined between two infinitely long horizontal coaxial cylinders can be built up by

using the laser-induced heating [10–12,14]. Indeed, since the laser beam can be focused to its diffraction limit, it can be used to inject energy at scales that are difficult to reach with other techniques [4].

In summary, we have reviewed the evolution of the director $\hat{\mathbf{n}}(t, \mathbf{r})$, velocity $\mathbf{v}(t, \mathbf{r})$, and temperature $T(t, \mathbf{r})$ in the hybrid aligned nematic cavity between two infinitely long coaxial cylinders to their equilibrium values, under influence of the radial temperature gradient directed from the cooler to warmer bounding surfaces. These calculations [28], based upon the classical Ericksen–Leslie theory, show that the microvolume of the HAN material under influence of the temperature gradient, settles down to a stationary flow regime in the horizontal direction. It has also been shown that the magnitude of that velocity $u_{eq}$ is proportional both to the tangential component of the stress tensor $\sigma_{zr}^{tm}$ and the size $d$ of the nematic cavity gap between two infinitely long coaxial cylinders, and the direction of $\mathbf{v} = u\hat{\mathbf{k}}$ influences both the direction of heat flux and the character of the preferred anchoring of the director to the bounding surfaces. These calculations also show that the optimal pumping effect in the microvolume HAN cavity between two coaxial cylinders build up under influence of the radial temperature gradient, when $a = \frac{R_1}{R_2 - R_1} \sim 0.01$, or when the radius of the homogeneously anchored warmer outward cylinder is greater than the radius of the homeotropically anchored cooler inward cylinder, approximately, in 100 times. Note that the above-mentioned flows are described by a continuous approach, where the manipulating fluid's length scales less than ten micrometers is reflected in the magnitudes of the parameters $\delta_i$ ($i = 6, 7, 8, 9$). It should be pointed out that the pumping effect under the influence of the radially directed thermal gradient is not found in the nematic cavity with the homeotropically or homogeneously aligned molecules at the restricted cylinders.

We believe that the present review can shed some light on the problem of the reorientation process in the microfibers under the influence of the temperature gradient [3,4,29]. We also believe that the precise handling of the liquid crystal microvolumes, which requires self-contained micropumps of small package size exhibiting a continuous volume flow, can be developed utilizing the interactions of both the director and velocity fields with the radially directed temperature gradient. Hence, the possible pumping technique described above appears applicable to various pumping strategies not involving mobile parts.

## 4. Conclusions

This review discusses some recent numerical advances in predicting the structural and hydrodynamic behavior of thermally excited vortical flows in microsized nematic channels caused by a laser beam focused both inside the nematic volume and on bounding surfaces. Despite the fact that certain qualitative and quantitative advances have been made in the hydrodynamic description of relaxation processes in microsized nematic volumes under the influence of a temperature gradient, there are still a number of questions concerning dissipative processes in confined liquid crystal phase with complex boundaries of nematic channels or with a free upper boundary of the liquid crystal material/air interface that need to be clarified.

It should be noted that in the case of a thick horizontal layer, with a thickness $d$ of several tens of millimeters, the Rayleigh–Benard instability may occur in a stationary liquid crystal volume heated from below. In that case, the LC volume is driven by maintaining the $\nabla T$, directed from the lower hotter boundary $T_{lw}$ to the upper cooler $T_{up}$ one. This instability occurs when the driving $\Delta T = T_{up} - T_{lw}$ is strong enough to overcome the dissipative effects of thermal conduction, thermomechanical effect, and viscosity. The control parameter describing the instability, the Rayleigh number $R = \alpha g d^3 \Delta T / (\alpha_4 / 2\rho)\kappa_{\perp}$, where $g$ is the gravitational acceleration, $\alpha$ is the isobaric thermal expansion coefficient, $\alpha_4 / 2\rho$ corresponds to the isotropic kinematic viscosity, and $\kappa_{\perp}$ is the perpendicular thermal diffusivity. The instability occurs at value $R = R_c \sim 1708$, independent of the fluid under consideration [30]. Taking into account that the size of the HAN channel $d \sim 5 - 10$ μm, in our case $R \ll R_c$, and the driving force is weak enough to set up of convection via the

Rayleigh–Benard mechanism [7]. Therefore, we are primarily concerned here with the description of the physical mechanism responsible for the horizontal flow in the HAN channel confined between two solid surfaces, which is excited by the temperature gradient $\nabla T$, and the magnitude of that flow $\mathbf{v}$ is $\sigma_{tm}$, whereas the direction of $\mathbf{v}$ is influenced both by the direction of the heat flux $\mathbf{q}$ and the character of the preferred anchoring of the average molecular direction $\hat{\mathbf{n}}$ to the restricted surfaces.

A number of suitable directions for the application of microliter LC materials are associated with thermally controlled manipulations in biology, medicine, and many areas of engineering [3–6]. This dynamic new field has also drawn attention due to advances in drug delivery [31,32]. There is another promising dynamic field of application of LC materials of microliter volumes. It is the artificial LC lenses with tunable focal length, which are called tunable lenses [33–35]. The idea of using a sessile drop of liquid as an optical lens dates back to the late 17th century. Over the last decade, researchers have become interested not only in liquid, but also in LC lenses due to exceptional properties of the LC/air interface [33,34]. In LC lenses, the focal length can be altered by varying the curvature of free LC/air interface in response to external stimuli or actuations. Taking into account the smoothness and flexibility of the LC/air interface, it is possible to solve a number of problems with optimizing these optical systems. For example, using laser-induced heating of microsized liquid crystal droplets, it is possible to manipulate the curvature of the LC/air interface and, as a result, the tunable focal length of lenses with curved surfaces. Another possible application of microfluidics is liquid crystal photonics [36].

Therefore, further study of a wider range of problems related to understanding how elastic soft matter, such as liquid crystals, begins to move under the influence of a temperature gradient, will require additional efforts, which will ultimately lead to an increase in our knowledge in the field of materials science. We also hope that this review will draw attention to the problem of manipulating the microsized volumes of liquid crystals by forming temperature gradients.

**Author Contributions:** I.Š.: writing—original draft preparation and editing. A.V.Z.: writing—original draft preparation and editing; supervision. All authors have read and agreed to the published version of the manuscript.

**Funding:** This research received no external funding.

**Data Availability Statement:** Not applicable.

**Conflicts of Interest:** The authors declare no conflict of interest.

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
