# Peer review of "Heat Driven Flows in Microsized Nematic Volumes: Computational Studies and Analysis"

_symmetry, doi:10.3390/sym13030459_

Round 1

Reviewer 1 Report

This is the review report on the nematic fluid pumping mechanism responsible for the heat driven flow in microfluidic nematic channels and capillaries, on the basis of the long-term work done by the author's group. The authors have already published many papers related to this topics, as that there  seems no weak point in the analyses and the computer simulation. This paper is helpful for readers to  understand the concept, analytical method, etc. As that the paper is worthy for publication. Nevertheless the paper is too  long and not easy to read everything.   To attract readers, the paper is necessary to be revised according to the following suggestions.

  1. This paper is very long, as that the key point and key conclusion should be summarized, before going into the detail description which appear in main sections, II,II, IV,
  2.  If there are  descriptions on new findings at the  initial part  of each section, e.g., section II, III, and IV, it will be very helpful for readers.  I can find the conclusions at the end of each sections, but I thought  it was easy to read  if I  could catch roughly  the findings, before reading the details.
  3.  The features of the vortex flow are  described in detail, and this is interesting, but  still not easy to understand the role of "heat".   If the authors can give more simple  physics image on the basis of analytical results, it will be helpful for readers.        

In summary, the paper is long but the paper will be acceptable after minor revision.

Author Response

The text of the manuscript has been modified according to referee’s suggestions. Our responses are given in Italic.

First of all, according to the first refere’s suggestion an additional figure was added to the text (now, the Fig.1, with explanation of the using geometry).

Second, the manuscript was made shorter. Now there are IV Sections and 39 figures.

Third, a number of references were added to the text instead of the previous ones.

This is the review report on the nematic fluid pumping mechanism responsible for the heat driven flow in microfluidic nematic channels and capillaries, on the basis of the long-term work done by the author's group. The authors have already published many papers related to this topics, as that there  seems no weak point in the analyses and the computer simulation. This paper is helpful for readers to  understand the concept, analytical method, etc. As that the paper is worthy for publication. Nevertheless the paper is too  long and not easy to read everything.   To attract readers, the paper is necessary to be revised according to the following suggestions.

  1. This paper is very long, as that the key point and key conclusion should be summarized, before going into the detail description which appear in main sections, II,II, IV,

As an introduction to each chapter, the necessary explanation of what will be set out has been provided.

  1.  If there are  descriptions on new findings at the  initial part  of each section, e.g., section II, III, and IV, it will be very helpful for readers.  I can find the conclusions at the end of each sections, but I thought  it was easy to read  if I  could catch roughly  the findings, before reading the details.
  2.  The features of the vortex flow are  described in detail, and this is interesting, but  still not easy to understand the role of "heat".   If the authors can give more simple  physics image on the basis of analytical results, it will be helpful for readers.   

   In the introduction to the paragraph 2.3 describing the formation of vortices under the influence of heat flux, an explanation is given due to what force the vortex arises. 

In summary, the paper is long but the paper will be acceptable after minor revision.

Reviewer 2 Report

The review of I. Sliwa and A. Zakharov is devoted to the theoretical study of heat driven flows in micro channels and capillaries. The study is based on the classical Ericksen-Leslie model with the corresponding heat terms. Certain mechanisms of temperature gradient formation was also discussed. The review is based mostly on the theoretical investigations of the nematic liquid crystal  (LC) flow under different geometries, which were published in Phys. Rev , J. Chem Phys and Phys Fluids by A. Zakharov et al. The review needs a major revision and resubmission due to the comments below.

  1. The review is full with formulas and drawings (44 Figs), which is very difficult to read. It looks more like a book, than a scientific review.
  2. is not complete, the authors mostly cited their own papers, despite the fact, that the subject was carefully discussed before (see e.g. https://yandex.ru/images/search?family=yes&pos=4&img_url=https%3A%2F%2Fwww.logobook.kz%2Fmake_nimage.php%3Fuid%3D13766308&text=chigrinov%20ooks&lr=11514&rpt=simage&source=wiz)
  3. There is absolutely no comparison of the proposed theory with the experimental results. Neither LC physical parameters, boundary conditions (anchoring energy) nor even thickness of the capillaries were not properly taken into account.
  4. The authors should consider the instabilities of the LC flow in the capillaries under certain conditions, which were properly described (see the Refs in item 2 above)
  5. LC microfluidic applications e.g. for sensors, lenses and other LC photonics devices should be also discussed (see e.g. https://yandex.ru/images/search?family=yes&pos=3&img_url=https%3A%2F%2Fnovapublishers.com%2Fwp-content%2Fuploads%2F2019%2F04%2F9781629483153-e1554137848866.jpg&text=chigrinov%20ooks&lr=11514&rpt=simage&source=wiz)

Author Response

The review of I. Sliwa and A. Zakharov is devoted to the theoretical study of heat driven flows in micro channels and capillaries. The study is based on the classical Ericksen-Leslie model with the corresponding heat terms. Certain mechanisms of temperature gradient formation was also discussed. The review is based mostly on the theoretical investigations of the nematic liquid crystal  (LC) flow under different geometries, which were published in Phys. Rev , J. Chem Phys and Phys Fluids by A. Zakharov et al. The review needs a major revision and resubmission due to the comments below.

First of all, according to the first referees’ suggestion an additional figure was added to the text (now, the Fig.1, with explanation of using geometry).

Second, the manuscript was made shorter. Now there are IV Sections and 39 figures.

Third, a number of references were added to the text instead of the previous ones.

  1. The review is full with formulas and drawings (44 Figs), which is very difficult to read. It looks more like a book, than a scientific review.

The manuscript was made shorter. Now there are IV Sections and 39 figures.

  1. is not complete, the authors mostly cited their own papers, despite the fact, that the subject was carefully discussed before (see e.g. https://yandex.ru/images/search?family=yes&pos=4&img_url=https%3A%2F%2Fwww.logobook.kz%2Fmake_nimage.php%3Fuid%3D13766308&text=chigrinov%20ooks&lr=11514&rpt=simage&source=wiz)

A number of references were added to the text instead of the previous ones.

  1. There is absolutely no comparison of the proposed theory with the experimental results.

At the moment, there are no experimentally obtained results concerning laser heating of microscopic volumes of nematic systems. There is only one publication concerning of circular flow formation triggered by Marangoni convection in nematic liquid crystal films with a free surface (see Ref.[10]). In this work the circular flow formation at a surface in homeotropically oriented nematic liquid crystals with a free surface using focused laser beam irradiation has been demonstrated.

  1. Neither LC physical parameters, boundary conditions (anchoring energy) nor even thickness of the capillaries were not properly taken into account.

All parameters of the LC system under consideration are corresponding to 4-cyano-4’-pentylbiphenyl~(5CB) (see page 12, above Eq.14.)

  1. The authors should consider the instabilities of the LC flow in the capillaries under certain conditions, which were properly described (see the Refs in item 2 above)
  2. LC microfluidic applications e.g. for sensors, lenses and other LC photonics devices should be also discussed (see e.g. https://yandex.ru/images/search?family=yes&pos=3&img_url=https%3A%2F%2Fnovapublishers.com%2Fwp-content%2Fuploads%2F2019%2F04%2F9781629483153-e1554137848866.jpg&text=chigrinov%20ooks&lr=11514&rpt=simage&source=wiz)

A number of suitable directions for the application of microliter LC materialsare discussed in the Conclusion section. One of these applications is related to the possibility of using microscopic droplets as LC lenses.

Reviewer 3 Report

The goal of this review paper is to summarize the main finding and understanding of heat-driven flow in micro-channels filled with nematic liquid crystals. These thermal effects are responsible for interesting phenomena such as the generation of vortices. The topic is interesting and I agree with the authors that a perspective on this topic is timely. 

However, I find this review paper to be way too focused on the authors' previous work. In fact, the first set of figures are all taken from the reference [21], while the following figures are taken from other recent work by the authors ([26, 11, 13]). This attitude is also clear by the fact that this long paper only has 35 citations and that 12 of them are of the last author. While I understand that the authors have done important work in this field and that it is only right that they should summarize in a review work, this should not be the sole focus. At the very least, the paper should contain either a perspective on how these simulations are informing recent experimental data, or a comparison with other theories in isotropic or liquid crystalline fluid. As it is, it is essentially a collection of previous papers by the same authors. For this reason, I think this paper cannot be published in its current form. 

Minor remarks: 

  • The choice of parameters delta on page 11 is not justified and should be discussed. In general, the model depend on several parameters but the parameters space is not sufficiently explored.
  •  I would advise against calling K31 the ratio K3/K1 because the symbol suggests a matrix element like K13 or K24. 
  • For the readability of the paper, i would recommend inserting a figure that represents the channel geometry and the choice of coordinates. Furthermore, in some figures the different lines are difficult to distinguish/follow (e.g. figure 2, figure 4)
  • The use of articles and prepositions in this paper are often incorrect and sentences are too long. I suggest extensive editing. 

Author Response

The goal of this review paper is to summarize the main finding and understanding of heat-driven flow in micro-channels filled with nematic liquid crystals. These thermal effects are responsible for interesting phenomena such as the generation of vortices. The topic is interesting and I agree with the authors that a perspective on this topic is timely. 

However, I find this review paper to be way too focused on the authors' previous work. In fact, the first set of figures are all taken from the reference [21], while the following figures are taken from other recent work by the authors ([26, 11, 13]). This attitude is also clear by the fact that this long paper only has 35 citations and that 12 of them are of the last author. While I understand that the authors have done important work in this field and that it is only right that they should summarize in a review work, this should not be the sole focus. At the very least, the paper should contain either a perspective on how these simulations are informing recent experimental data, or a comparison with other theories in isotropic or liquid crystalline fluid. As it is, it is essentially a collection of previous papers by the same authors. For this reason, I think this paper cannot be published in its current form. 

First of all, according to the first referees’ suggestion an additional figure was added to the text (now, the Fig.1, with explanation of using geometry).

Second, the manuscript was made shorter. Now there are IV Sections and 39 figures.

Third, a number of references were added to the text instead of the previous ones.

Minor remarks: 

  • The choice of parameters delta on page 11 is not justified and should be discussed. In general, the model depend on several parameters but the parameters space is not sufficiently explored.

For the case of 4cyano4pentylbiphenyl (5CB), at temperature corresponding to

nematic phase, the first four parameters δ1, δ2 , δ3, and δ4, that are involved in Eqs.(6)-(8) has been defined.

  • I would advise against calling K31 the ratio K3/K1 because the symbol suggests a matrix element like K13 or K24. 

It was corrected. Now the ratio K3/K1 is defined as K. The same used for g=g2/g1.

  • For the readability of the paper, i would recommend inserting a figure that represents the channel geometry and the choice of coordinates. Furthermore, in some figures the different lines are difficult to distinguish/follow (e.g. figure 2, figure 4)

It was done (see Fig.1)

  • The use of articles and prepositions in this paper are often incorrect and sentences are too long. I suggest extensive editing. 

It was done too.

Reviewer 4 Report

The present manuscript by I. Sliwa and A. Zakharov certainly covers a topic of much interest in the area of heat driven flows of nematic liquid crystals. A review on this topic is thus a welcome contribution to the literature, being timely and of applicational relevance for scientists.

My main reluctance with respect to this contribution is the fact that this is not really a review. It is rather a summary and collation of already published own works, all written by the same group of authors (Zakharov, A. V.; Vakulenko, A. A.). A review should at least put one’s own work into context with other worldwide ongoing research and activities on the topic. Furthermore, 71 p. of the text with many formulas make this article not attractive to potential readers.

A second issue is English language, terminology (e.g., "Slow heating regime when a laser beam is focused on a lower boundary" I would formulate as "Slow heating mode with the focused laser beam on the bottom substrate" (?)) and grammar, which would greatly be improved in proof-reading by a native speaker.

Author Response

The present manuscript by I. Sliwa and A. Zakharov certainly covers a topic of much interest in the area of heat driven flows of nematic liquid crystals. A review on this topic is thus a welcome contribution to the literature, being timely and of applicational relevance for scientists.

First of all, according to the first referee’s suggestion, an additional figure was added to the text (now, the Fig.1, with explanation of using geometry).

Second, the manuscript was made shorter. Now there are IV Sections and 39 figures.

Third, a number of references were added to the text instead of the previous ones.

My main reluctance with respect to this contribution is the fact that this is not really a review. It is rather a summary and collation of already published own works, all written by the same group of authors (Zakharov, A. V.; Vakulenko, A. A.). A review should at least put one’s own work into context with other worldwide ongoing research and activities on the topic. Furthermore, 71 p. of the text with many formulas make this article not attractive to potential readers.

The manuscript was made shorter. Now there are IV Sections and 39 figures.

A second issue is English language, terminology (e.g., "Slow heating regime when a laser beam is focused on a lower boundary" I would formulate as "Slow heating mode with the focused laser beam on the bottom substrate" (?))

Throughout the text, the “slow heating regime” has been replaced with the “slow heating mode”.

and grammar, which would greatly be improved in proof-reading by a native speaker.

It was done.

Round 2

Reviewer 2 Report

The authors partially answered my questions. But the new version looks very similar to the previous one. I do believe the authors should consider my review more carefully. I do not think that the present version is acceptable 

Author Response

The authors have repeatedly noted in the text that to date there is a small number of works devoted to the study of the influence of heat gradients on the formation of hydrodynamic flows in microfluidic nematic channels and capillaries. Most of the progress was achieved with the help of numerical studies carried out in the framework of hydrodynamic models described in the review. The small number of works that were devoted to the above-described subject of analysis are given in the review. Since there are currently no experimental works, the theoretical results presented by the authors can serve as guidelines for future experiments.

Returning to the problem of instability of the hydrodynamic flow formed under the action of a thermomechanical force, it should be noted that since there are no external factors that would distabilize the hydrodynamic flow, it makes no sense to talk about instability. The stability of the numerical procedure for equations (6), (14), and (8) is discussed in detail on page 12.

Reviewer 3 Report

The authors of the review have added figure 1 in their revision and they have shortened the manuscript by essentially eliminating section IV - which was not requested by me. However, they have not addressed any of the major concerns I have expressed previously, therefore my opinion is unchanged. I think that the review scope is too narrow and that it only describes some of the (important and worthwhile) work of the authors that can be easily accessed from their original papers. Again I want to remark that this is not a criticism of the work of the authors but just of the scope of their review paper. 

Author Response

The authors have repeatedly noted in the introduction section as well in the sec. II, that to date there is a small number of works devoted to the study of the influence of heat gradients on the formation of hydrodynamic flows in microfluidic nematic channels and capillaries. Most of the progress was achieved with the help of numerical studies carried out in the framework of hydrodynamic models described in the review.

In Chapter II, we point out that “So, we are primary concerned here with the description of the physical mechanism responsible for the flow v in the LC channel confined between solid surfaces, which is excited by the temperature gradient ÑT, and the magnitude of that flow is proportional to DT, whereas the direction of v influences both the direction of the heat flux q and the character of the preferred anchoring of the average molecular direction n to the restricted surfaces”. We focused on the analysis of the problem of formation of hydrodynamic flows v in microsized nematic channels under the influence of directed heat fluxes from the position of numerical methods. The analysis of the dynamics of Lehmann-type effects in chiral LCs is far from our research interests. For more information see, for instance, the Ref.[22].

In an attempt to better understand the dissipation  processes in microfluidic nematic systems confined in the various microsized geometries, under the influences the temperature gradient, we will review a number of numerical studies of these nematic systems. With this in aim, a number of theories which include the hydrodynamic equations describing both the director reorientation and flow of nematic fluid, as well as the redistribution of the temperature field, will be described. These equations also  will be supplemented by appropriate anchoring conditions for the director field on the bounding surfaces, as well as the no-slip condition for the velocity field.

Since the geometry of the microsized nematic channel or capillary affects the nature of the hydrodynamic flow that is formed, two cases will be considered. The first is when dealing with a rectangular channel, and second is the cylindrical capillary.”

 The small number of works that were devoted to the above-described subject of analysis are given in the review. Since there are currently no experimental works, the theoretical results presented by the authors can serve as guidelines for future experiments.

Reviewer 4 Report

Not to mention that I have rated the manuscript well, my opinion is that the paper will be very long. As the result, it might be not interesting to potential readers. Moderate English changes are still required, but this can be done during proof reading.

Author Response

The authors have repeatedly noted in the introduction section as well in the sec. II, that to date there is a small number of works devoted to the study of the influence of heat gradients on the formation of hydrodynamic flows in microfluidic nematic channels and capillaries. Most of the progress was achieved with the help of numerical studies carried out in the framework of hydrodynamic models described in the review.

In this review, we try to convey to the readers the main problems that researchers face when studying the effect of temperature gradients on the process of reorientation of the director field, velocity field, and temperature in microscopic nematic channels and capillaries. And since we cover this problem in the framework of a nonlinear generalization of the Ericksen-Leslie theory, the fact that you have to deal with complex analytical and computational procedures is the price that you have to pay to achieve the result. The reader will decide for himself whether he should dive into this analysis, or limit himself only to the final results.

Round 3

Reviewer 2 Report

The authors still did not take properly my comments into account. As for hydrodynamic instabilities  under the heat of liquid crystals, this is a well known issue and there are lots of related experimental materials (see e.g. https://www.amazon.com/Electrooptic-Effects-Crystal-Materials-Partially/dp/0387940308/ref=sr_1_2?dchild=1&qid=1614570297&refinements=p_27%3AV.+G.+Chigrinov&s=books&sr=1-2)

There is also very few comments on the applications of the microfluidic flow in LC photonics (see e.g. https://www.amazon.com/Liquid-Crystal-Photonics-Engineering-Techniques/dp/162948315X/ref=sr_1_4?dchild=1&qid=1614570542&refinements=p_27%3AVladimir+G.+Chigrinov&s=books&sr=1-4)

All the review is the general theory with a lot of the details and equations, which will be difficult to be used for the researchers who may like to make experimental verification of the proposed theory and/or used for the applications in sensors, lenses or other LC photonics devices. I do hope, that the authors will take my comments into account.

Author Response

The text of the manuscript has been modified according to referee’s suggestions. Our responses are given in Italic.

First of all, according to the referee’s suggestion an additional comments were added to the text (now, to the Conclusion Sec., with discussion about the possible Rayleigh-Benard instability.

Second, an extra two References were added to the text (now Refs. [30] and [36]).

The authors still did not take properly my comments into account. As for hydrodynamic instabilities  under the heat of liquid crystals, this is a well known issue and there are lots of related experimental materials (see e.g. https://www.amazon.com/Electrooptic-Effects-Crystal-Materials-Partially/dp/0387940308/ref=sr_1_2?dchild=1&qid=1614570297&refinements=p_27%3AV.+G.+Chigrinov&s=books&sr=1-2)

In the framework of the hydrodynamic problems described above, only one instability can occur in a thick layer of a liquid crystal. This is the Rayleigh-Benard instability. In this case of the thick horizontal layer with a thickness d of several tens of millimeters, the Rayleigh-Benard  instability may occur in a stationary liquid crystal volume heated from below. The LC volume is driven by maintaining the ÑT, directed from the lower surface at a temperature Tlw to the upper surface temperature Tup (Tlw> Tup). For small driving the fluid remains at rest, and a linear temperature profile is set up across the LC sample. Due to the thermal expansion, however, the fluid near the lower restricted surface is less dense and an intrinsically instable in the gravitational field. This instability occurs when the driving DT is strong enough to overcome the dissipative effects of thermal conduction, thermomechanical effect, and viscosity. The control parameter describing the instability, the Rayleigh number R=2ragd3DT/(a4k^), where g is the gravitational acceleration, a is the isobaric thermal expansion coefficient, a4/(2r) corresponds to the isotropic kinematic viscosity, and k^ is the perpendicular thermal diffusivity. The instability occurs at the value R=Rc~ 1708, independent of the fluid under consideration. Taking into account that the size of the LC cell d~5-10 mm, in our case R<<Rc, and the driving force is weak enough to set up of convection via the Rayleigh-Benard mechanism. So, we are primary concerned here with the description of the physical mechanism responsible for the horizontal flow in the LC phase confined between two solid surfaces, which is excited by the temperature gradient ÑT, and the magnitude of that flow is ~stm, whereas the direction of v is influenced both by the direction of the heat flux q and the character of the preferred anchoring of the average molecular direction n  to the restricted surfaces.

There is also very few comments on the applications of the microfluidic flow in LC photonics (see e.g. https://www.amazon.com/Liquid-Crystal-Photonics-Engineering-Techniques/dp/162948315X/ref=sr_1_4?dchild=1&qid=1614570542&refinements=p_27%3AVladimir+G.+Chigrinov&s=books&sr=1-4)

In the Conclusion Section, we pointed to the fact that microfluidic of liquid crystals also finds application in LC Photonics (see Ref.[36]).

All the review is the general theory with a lot of the details and equations, which will be difficult to be used for the researchers who may like to make experimental verification of the proposed theory and/or used for the applications in sensors, lenses or other LC photonics devices. I do hope, that the authors will take my comments into account.

In the Conclusion Section, we commented the fact that the final analytical expressions may cause some difficulties in the process of their use by experimentalists.  But we hope that this review will draw attention to the problem of manipulating the micro-dimensional volumes of liquid crystals by forming temperature gradients.

All the referee's comments were addressed.

We hope that the manuscript in the present form fulfills the requirements for publication in the Symmetry.
